# Deep Metric Learning with Spherical Embedding

**Dingyi Zhang**[1], **Yingming Li**[1*], **Zhongfei Zhang**[2]
[1]College of Information Science & Electronic Engineering, Zhejiang University, China
[2]Department of Computer Science, Binghamton University, USA
{dyzhang, yingming}@zju.edu.cn, zhongfei_mark@yahoo.com

## Abstract

Deep metric learning has attracted much attention in recent years, due to seamlessly combining the distance metric learning and deep neural network. Many endeavors are devoted to design different pair-based angular loss functions, which decouple the magnitude and direction information for embedding vectors and ensure the training and testing measure consistency. However, these traditional angular losses cannot guarantee that all the sample embeddings are on the surface of the same hypersphere during the training stage, which would result in unstable gradient in batch optimization and may influence the quick convergence of the embedding learning. In this paper, we first investigate the effect of the embedding norm for deep metric learning with angular distance, and then propose a spherical embedding constraint (SEC) to regularize the distribution of the norms. SEC adaptively adjusts the embeddings to fall on the same hypersphere and performs more balanced direction update. Extensive experiments on deep metric learning, face recognition, and contrastive self-supervised learning show that the SEC-based angular space learning strategy significantly improves the performance of the state-of-the-art.

## 1  Introduction

The objective of distance metric learning is to learn an embedding space where semantically similar instances are encouraged to be closer than semantically different instances [1, 2]. In recent years, with the development of deep learning, deep metric learning (DML) demonstrates evident improvements by employing a neural network as the embedding mapping. With an appropriate distance metric, it is convenient to handle many visual understanding tasks, such as face recognition [3, 4] and fine-grained image retrieval [5, 6, 7]. In regard to the research in DML, an active direction is to design a discriminative loss function for model optimization. A family of pair-based loss functions are proposed, which are constructed by similarities of instance pairs in a mini-batch, such as contrastive loss [8], triplet loss [9, 4], lifted structured loss [5], $N$-pair loss [10], and multi-similarity loss [11].

Theoretically, either Euclidean distance or angular distance could be employed to measure the similarity between two embeddings in an embedding space, while in the existing DML loss functions, angular distance is usually adopted to disentangle the norm and direction of an embedding, which ensures the training and testing measure consistency. However, this traditional setup usually ignores the importance of the embedding norm for gradient computation. For example, considering a cosine distance which measures the angular distance between two embeddings $f_i$ and $f_j$, its gradient to an embedding $f_i$ is computed as follows:

$$\frac{\partial \langle \hat{f}_i, \hat{f}_j \rangle}{\partial f_i} = \frac{1}{||f_i||_2}(\hat{f}_j - \cos \theta_{ij} \hat{f}_i), \tag{1}$$

where $\hat{f}$ denotes the $l_2$-normalized embedding of $f$.

---

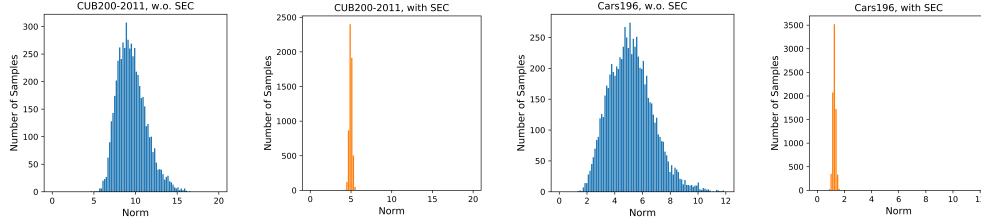

Figure 1: We train models on CUB200-2011 and Cars196 datasets, all with a triplet loss using a cosine distance. Here we show the distribution of learned embeddings' norms with and without SEC.

From the above gradient analysis, we see that the embedding norm plays an important role in the gradient magnitude, which is also mentioned in [12]. When angular distances are optimized in the loss function, it requires the embeddings to have similar norms to achieve a balanced direction update. However, most existing angular loss-based methods cannot guarantee that all the sample embeddings are on the surface of the same hypersphere during the training stage. As shown in Figure 1, the distribution of learned embeddings' norms with the angular triplet loss have a large variance in the training stage. Consequently, the gradient correspondingly would become unstable in batch optimization and may influence the quick convergence of the embedding learning. For example, the direction updating is relatively slower for embeddings with larger norms.

To address the above limitation, in this paper, we propose a spherical embedding constraint (SEC) method for better embedding optimization. SEC attempts to make the embeddings to fall on the surface of the same hypersphere by adaptively adjusting the norms of embeddings. Instead of directly constraining these norms to be a fixed value, which represents the radius of the hypersphere, SEC flexibly reduces the variance of the embedding norm and constrains these norms to be similar. During training, SEC operates on each mini-batch and all embedding norms are pushed to their average value. As shown in Figure 1, the variance of the embedding norms are reduced so that a balanced direction update is performed. Extensive evaluations are conducted on deep metric learning, face recognition, and contrastive self-supervised learning to investigate the performance of angular space learning with SEC. The experiment results on several public datasets show that the proposed method significantly improves the performance of the state-of-the-art.

## 2 Related work

**Batch normalization.** Ioffe and Szegedy propose batch normalization (BN) method [13] to deal with the change of input distribution of layers in CNNs. This method has been shown to be effective to accelerate the convergence and to enhance the generalization ability of CNNs. By inserting a BN layer into an arbitrary position in CNNs (usually before the nonlinear function), a Gaussian-like distribution at each dimension of the output is expected to be obtained in a mini-batch. Our SEC also attempts to perform an operation similar to this normalization for obtaining a better generalization performance. The difference is that SEC focuses on optimizing embeddings in an angular space and only restricts the norms of final embeddings to be similar to make them on the same hypersphere.

**Angular distance optimization in pair-based and classification loss functions.** In deep metric learning task, much effort has been devoted to design pair-based loss functions. Triplet loss [9, 4] encourages the distance of a negative pair to be larger than that of a positive pair by a given margin. $N$-pair loss [10] extends the triplet loss and pushes more than one negative samples farther away from the anchor simultaneously compared with the positive sample. Multi-similarity loss [11] considers both self-similarity and relative similarity for weighting informative pairs by two iterative steps. Other loss functions includes lifted structured loss [5], proxy-NCA [14], clustering [15], hierarchical triplet loss [16], ranked list loss [17], and tuplet margin loss [18]. Among these methods, angular distance optimization has become a common approach and is employed by most of loss functions mentioned above. With this setup, they decouple the magnitude and direction information of embedding vectors and aim to optimize the angular distance between two embeddings. This way dose achieve a better performance, guaranteeing the consistent training and testing measurement. On the other hand, in face recognition task, researchers also find that in softmax loss, replacing the inner product between weight vectors and embeddings by cosine distance provides better results. A series of cosine-based softmax loss functions have gradually been proposed, including $l_2$-softmax [22], normface [23], sphereface [24], cosface [25], and arcface [26]. In addition, recent contrastive learning algorithms

for self-supervised learning also adopt the embedding normalization step and attempt to maximize the cosine similarity between two embeddings generated from a positive pair, *i.e.*, two differently augmented versions or two different views of the same image, with contrastive losses, such as SimCLR [19], CMC [20], and [21]. However, the traditional angular loss setup cannot guarantee that all the sample embeddings are on the surface of the same hypersphere during the training stage, which is usually ignored by the current methods. In this paper, we first investigate the importance of the embedding norm to direction update in batch optimization and then introduce the SEC to improve the optimization process. Further, SEC attempts to perform more balanced embedding update by adaptively adjusting the norms for embeddings and is complementary to the above loss functions.

## 3 Method

### 3.1 Revisiting pair-based angular loss functions for deep metric learning

Suppose that we are provided with a set of training images of $K$ classes. We first extract the feature embedding of each sample by CNN and obtain $\{(f, y), \cdots\}$, where $f \in \mathcal{R}^D$ denotes the feature embedding and $y \in \{1, \cdots, K\}$ is the corresponding label. A normalized Euclidean distance or a cosine distance is usually employed to measure the similarity between two embeddings $f_i$ and $f_j$,

$$\text{normalized Euclidean distance: } S_{ij} = ||\hat{f}_i - \hat{f}_j||_2^2$$

$$\text{cosine distance: } S_{ij} = \langle \hat{f}_i, \hat{f}_j \rangle$$

where $\hat{f}$ denotes the $l_2$-normalized embedding with a unit norm from the original embedding $f$, *i.e.*, $\hat{f} = \frac{f}{||f||_2}$. The above two measures are equivalent for computing the angular distance between two embeddings, since $||\hat{f}_i - \hat{f}_j||_2^2 = 2 - 2\langle \hat{f}_i, \hat{f}_j \rangle$.

Then different pair-based loss functions can be constructed by the above similarities. Let $S_{ap}$ denote the similarity of positive pair $(\hat{f}_a, \hat{f}_p)$ and $S_{an}$ denote the similarity of negative pair $(\hat{f}_a, \hat{f}_n)$, where labels satisfy $y_a = y_p \neq y_n$. The classical triplet loss [4] and tuplet loss (also refereed as normalized $N$-pair loss by us) [10, 18] can be formulated as

$$L_{\text{triplet}} = (||\hat{f}_a - \hat{f}_p||_2^2 - ||\hat{f}_a - \hat{f}_n||_2^2 + m)_+ \tag{2}$$

$$L_{\text{tuplet}} = \log[1 + \sum_n e^{s(\langle \hat{f}_a, \hat{f}_n \rangle - \langle \hat{f}_a, \hat{f}_p \rangle)}], \tag{3}$$

where $m$ is a margin hyper-parameter and $s$ is a scale hyper-parameter. Both of them optimize the embeddings in an angular space.

**Existing problem.** Though the angular space learning ensures the training and testing measure consistency by decoupling the norm and direction of an embedding, the existing pair-based loss functions for angular distance optimization usually ignore the importance of the embedding norm distribution during the training stage. As shown in Figure 1, the distribution of learned embeddings' norms with the vanilla angular triplet loss has a large variance in the training stage, which means that the embeddings are not on the surface of the same hypersphere, resulting in unbalanced direction update for different embeddings and influencing the stability of batch optimization.

### 3.2 The effect of embedding norm to the optimization process

In this part, the effect of embedding norm is investigated for the optimization of the existing pair-based angular loss functions. We draw two conclusions: (1) when the angular distances are optimized in these loss functions, the gradient of an embedding is always orthogonal to itself. (2) the direction updating of the embedding is easy to be influenced by the large variance norm distribution, resulting in unstable batch optimization and a slower convergence rate.

Since gradient descent-based methods are mainly adopted for deep model optimization, here we would analyze the effect of the embedding norm from the perspective of the gradient. Considering a pair-based angular loss function $L$, its gradient to an embedding $f_i$ can be formulated as

$$\frac{\partial L}{\partial f_i} = \sum_{(i,j)} \frac{\partial L}{\partial S_{ij}} \frac{\partial S_{ij}}{\partial f_i} = \sum_{(i,j)} \phi_{ij} \frac{\partial S_{ij}}{\partial f_i} = \sum_{(i,j)} \phi_{ij} \frac{\kappa}{||f_i||_2} (-\hat{f}_j + \cos\theta_{ij}\hat{f}_i), \tag{4}$$

where $\kappa = 2$ if $L$ adopts a normalized Euclidean distance and $\kappa = -1$ with a cosine distance. Here $(i, j)$ is a positive or negative index pair, $\cos\theta_{ij} = \langle \hat{f}_i, \hat{f}_j \rangle$ denotes the cosine distance between

embeddings $f_i$ and $f_j$, and $\phi_{ij} = \frac{\partial L}{\partial S_{ij}}$ is a scalar function which is only related to all $S_{ik}$ terms in the loss function, where $S_{ik}$ could be seen as the angular relationship between embeddings $f_i$ and $f_k$ and $k$ is the index of a positive or negative sample.

**Proposition 1.** *For a pair-based angular loss function, i.e., it is constructed by similarities measured with a normalized Euclidean distance or cosine distance, then its gradient to an embedding is orthogonal to this embedding, i.e., $\langle f_i, \frac{\partial L}{\partial f_i} \rangle = 0$.*

*Proof.* Based on Equation 4, we calculate the inner product between an embedding and the gradient of the loss function to it, *i.e.*,

$$\langle f_i, \frac{\partial L}{\partial f_i} \rangle = \sum_{(i,j)} \phi_{ij} \frac{\kappa}{||f_i||_2} \langle f_i, (-\hat{f}_j + \cos\theta_{ij}\hat{f}_i) \rangle$$

$$= \sum_{(i,j)} \phi_{ij} \frac{\kappa}{||f_i||_2} (-||f_i||_2 \cos\theta_{ij} + \cos\theta_{ij}||f_i||_2) = 0, \quad (5)$$

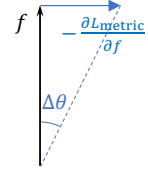

This conclusion is similar to that in [23]. From Proposition 1, since the gradient of a specific embedding is orthogonal to itself, the change of its direction at each update would be conveniently calculated, as shown in Figure 2. Here the tangent of the angular variation $\Delta\theta$ is used to measure the magnitude of direction change, as tangent is a monotonically increasing function when the angle is in $[0, \pi/2)$. Consequently, for embedding $f_i$, we have

$$\tan(\Delta\theta)_i = ||\frac{\partial L}{\partial f_i}||_2/||f_i||_2 = \frac{1}{||f_i||_2^2}||\sum_{(i,j)} \phi_{ij}\kappa(-\hat{f}_j + \cos\theta_{ij}\hat{f}_i)||_2. \quad (6)$$

Figure 2: An illustration of $\Delta\theta$.

**Proposition 2.** *Considering a pair-based angular loss function, we assume that the angular relationships among embeddings are fixed. For a specific embedding at one update, if it has a larger norm, then it gets a smaller change in its direction and vice versa.*

*Proof.* From Equation 6, we observe that $\frac{1}{||f_i||_2^2}$ is the only term which is related to the norm of this embedding, while the other terms are constants or only related to the angular relationship among embeddings. Therefore, when

$$||f_i^{(1)}||_2 > ||f_i^{(2)}||_2, \quad (7)$$

we obtain

$$\tan(\Delta\theta)_i^{(1)} < \tan(\Delta\theta)_i^{(2)}, \quad (8)$$

which indicates a smaller change in this embedding's direction updating.

From Proposition 2, for a specific embedding, the change in its direction at one update is not only related to the angular relationship, but also inversely proportional to the square of its norm. A similar observation is also reported in [12]. However, we note that the above conclusion is only based on the vanilla SGD method and next we also attempt to explain how other optimizers affect the direction update. We start with expressing the above conclusion of the vanilla SGD method more formally as below, where with the learning rate $\alpha$, an embedding is updated at the $t$-th iteration by

$$f_{t+1} = f_t - \alpha \frac{\partial L}{\partial f_t}. \quad (9)$$

**Proposition 3.** *With vanilla SGD, the embedding direction is updated by*

$$\hat{f}_{t+1} = \hat{f}_t - \frac{\alpha}{||f_t||_2^2}(I - \hat{f}_t\hat{f}_t^\top)\frac{\partial L}{\partial \hat{f}_t} + O(\alpha^2). \quad (10)$$

*Proof.* We first rewrite Equation 4 without the subscript as

$$\frac{\partial L}{\partial f} = (\frac{\partial \hat{f}}{\partial f})^\top \frac{\partial L}{\partial \hat{f}} = \frac{1}{||f||_2}(I - \hat{f}\hat{f}^\top)\frac{\partial L}{\partial \hat{f}}. \quad (11)$$

Then based on the above equation and Equation 9, we have

$$||f_{t+1}||_2^2 = ||f_t||_2^2 + \frac{\alpha^2}{||f_t||_2^2}[(I - \hat{f}_t\hat{f}_t^\top)\frac{\partial L}{\partial \hat{f}_t}]^\top[(I - \hat{f}_t\hat{f}_t^\top)\frac{\partial L}{\partial \hat{f}_t}],$$

and thus

$$||f_{t+1}||_2 = \sqrt{||f_t||_2^2[1 + \frac{\alpha^2}{||f_t||_2^4}(\frac{\partial L}{\partial \hat{f}_t})^\top(I - \hat{f}_t\hat{f}_t^\top)\frac{\partial L}{\partial \hat{f}_t}]} = ||f_t||_2 + O(\alpha^2).$$

Besides, from Equation 9 and 11 we also have

$$||f_{t+1}||_2 \hat{f}_{t+1} = ||f_t||_2 \hat{f}_t - \frac{\alpha}{||f_t||_2}(I - \hat{f}_t \hat{f}_t^\top)\frac{\partial L}{\partial \hat{f}_t}.$$

Finally we combine the above results and obtain

$$\hat{f}_{t+1} = \frac{||f_t||_2}{||f_{t+1}||_2}\hat{f}_t - \frac{\alpha}{||f_{t+1}||_2 ||f_t||_2}(I - \hat{f}_t \hat{f}_t^\top)\frac{\partial L}{\partial \hat{f}_t} = \hat{f}_t - \frac{\alpha}{||f_t||_2^2}(I - \hat{f}_t \hat{f}_t^\top)\frac{\partial L}{\partial \hat{f}_t} + O(\alpha^2).$$

Then we consider SGD with momentum method, which updates the embedding by

$$v_{t+1} = \beta v_t + \frac{\partial L}{\partial f_t}, f_{t+1} = f_t - \alpha v_{t+1}, \tag{12}$$

and Adam method [27], which updates the embedding by

$$v_{t+1} = \beta_1 v_t + (1 - \beta_1)\frac{\partial L}{\partial f_t}, g_{t+1} = \beta_2 g_t + (1 - \beta_2)||\frac{\partial L}{\partial f_t}||_2^2, f_{t+1} = f_t - \alpha\frac{v_{t+1}/(1 - \beta_1^t)}{\sqrt{g_{t+1}/(1 - \beta_2^t)} + \epsilon}. \tag{13}$$

**Proposition 4.** *When using SGD with momentum, the embedding direction is updated by*

$$\hat{f}_{t+1} = \hat{f}_t - \frac{\alpha}{||f_t||_2^2}(I - \hat{f}_t \hat{f}_t^\top)[||f_t||_2 \beta v_t + (I - \hat{f}_t \hat{f}_t^\top)\frac{\partial L}{\partial \hat{f}_t}] + O(\alpha^2). \tag{14}$$

**Proposition 5.** *With Adam, the embedding direction is updated by*

$$\hat{f}_{t+1} = \hat{f}_t - \frac{\alpha}{||f_t||_2}(I - \hat{f}_t \hat{f}_t^\top)\frac{\sqrt{1 - \beta_2^t}[||f_t||_2 \beta_1 v_t + (1 - \beta_1)(I - \hat{f}_t \hat{f}_t^\top)\frac{\partial L}{\partial \hat{f}_t}]}{(1 - \beta_1^t)\sqrt{||f_t||_2^2 \beta_2 g_t + (1 - \beta_2)(\frac{\partial L}{\partial \hat{f}_t})^\top(I - \hat{f}_t \hat{f}_t^\top)\frac{\partial L}{\partial \hat{f}_t}}} + O(\alpha^2). \tag{15}$$

The proofs are provided in Appendix A. From the above propositions, with a small global learning rate $\alpha$, $\frac{\alpha}{||f_t||_2^2}$ and $\frac{\alpha}{||f_t||_2}$ could be approximately seen as the effective learning rate for updating the embedding direction with vanilla SGD (SGD with momentum) and Adam method, respectively. Thus, with different optimizers, the embedding norm would always play an important role in the direction updating. With a large norm, an embedding may update slowly and get stuck within a small area in the embedding space. On the other hand, if the norm is too small, then this embedding may take a quite large step at one update. Due to this effect, similar norms of embeddings would be more preferable to attaining more balanced direction update. However, we actually observe a large variance from the norm distribution when learning with a pair-based angular loss function on different datasets, as shown in Figure 1. It shows that the traditional angular loss setup cannot guarantee that the learned embeddings lie on the surface of the same hypersphere. Consequently, the gradient would become unstable in batch optimization during training, due to unbalanced direction update among embeddings, which slows the convergence rate and degrades the generalization performance.

### 3.3  Spherical embedding learning

Based on the above analysis, since the large variance of norms would make embeddings suffering from unbalanced direction update, it is necessary to constrain the embedding norm during training to eliminate this negative effect. One straightforward idea is to attempt to alleviate this problem by constraining the embeddings to lie on the surface of the same hypersphere so that they have the identical norm. Mathematically, it could be formulated as a constrained optimization problem:

$$\min_{\vartheta} L(\{(f_i, y_i)\}_{i=1}^N; \vartheta) \quad \text{s.t. } \forall i, \; ||f_i||_2 = \mu, \tag{16}$$

where $L$ is a pair-based angular loss function, $\vartheta$ is the model parameter, and $\mu$ is the radius of the hypersphere. With a quadratic penalty method, this problem could be easily transformed to an unconstrained one as follows:

$$\min_{\vartheta} L(\{(f_i, y_i)\}_{i=1}^N; \vartheta) + \eta * \frac{1}{N}\sum_{i=1}^N (||f_i||_2 - \mu)^2, \tag{17}$$

where $\eta$ is a penalty weight for the norm constrain. However, to solve this problem, we still need to determine the value of the hyper-parameter $\mu$, which is inconvenient for training. Instead, we consider a parameter-free scheme, where $\mu$ is decided by the average norm of all embeddings, *i.e.*, $\mu = \frac{1}{N}\sum_{j=1}^N ||f_j||_2$, and is calculated in each mini-batch in practice. Further, the second term in

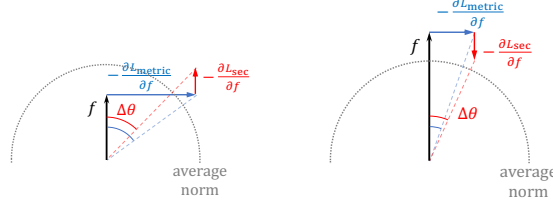

Figure 3: SEC adaptively adjusts the direction update $\Delta\theta$ at each iteration by adding a new gradient component $\frac{\partial L_{\text{sec}}}{\partial f}$ to $\frac{\partial L_{\text{metric}}}{\partial f}$.

Equation 17 is named as a *spherical embedding constraint* (SEC), *i.e.*, $L_{\text{sec}} = \frac{1}{N}\sum_{i=1}^{N}(||f_i||_2 - \mu)^2$. During training, the complete objective function is $L = L_{\text{metric}} + \eta * L_{\text{sec}}$. With SEC, two benefits are obtained: (1) it would alleviate unbalanced direction update by adaptively increasing (decreasing) the change of direction for embeddings with large (small) norms at one update. (2) the change of an embedding's direction is almost only related to its angular relationships with the other embeddings.

In Figure 3, we illustrate how SEC adjusts the direction update for different embeddings. Considering the gradient of it to an embedding, i.e.,

$$\frac{\partial L_{\text{sec}}}{\partial f_i} = \frac{2}{N}(||f_i||_2 - \mu)\hat{f}_i. \tag{18}$$

From the above equation, SEC provides an update direction which is parallel to the embedding. Consequently, for an embedding whose norm is smaller (larger) than the average norm, SEC attempts to increase (decrease) its norm at the current update. With this newly added gradient component, the total angular change of an embedding is also adjusted, as shown in Figure 3. For embeddings with different norms, the angular changes they obtain would be less influenced by their norms than without SEC. It thus leads to more balanced direction updating for embeddings especially when the norms have an extremely large variance. Besides, SEC gradually narrows the norm gap at each iteration and with the variance becoming smaller and smaller, this negative effect is further eliminated.

When the variance becomes relatively small, *i.e.*, different embeddings almost locate on the same hypersphere, from Equation 4 it is shown that the magnitude of the gradient is almost only determined by the angular relationship among embeddings, *i.e.*, $\frac{\partial L}{\partial S_{ij}}$. From the general pair weighting framework in [11], for a pair-based loss function, $\frac{\partial L}{\partial S_{ij}}$ can be seen as the weight assigned to $S_{ij}$. Different pair-based loss functions are designed to assign the required weights for harder pairs. During the training stage, since hard pairs are difficult to learn, the larger gradients are obtained to encourage the model to pay more attention to them, which implicitly benefits the model performance. Overall, the magnitude of the gradient plays an important role in embedding optimization in an angular space.

In addition to SEC, here we also discuss another norm regularization method proposed in [10], which aims to regularize the $l_2$-norm of embeddings to be small and is refereed as $L^2$-reg by us. This method could be seen as a special case of SEC with $\mu = 0$ and the comprehensive comparisons between them are provided in Section 4. It is shown that SEC is more favorable than $L^2$-reg, indicating a better way for norm distribution adjustment during training.

### 3.4 An extension to cosine-based softmax losses and contrastive losses

Recently, cosine-based softmax losses, such as normface [23], sphereface [24], cosface [25], and arcface [26], have achieved a remarkable performance in face recognition task. For example, the loss function of cosface is formulated as follows:

$$L = -\log\frac{e^{sS_{i,y_i}}}{e^{sS_{i,y_i}} + \sum_{j\neq y_i}e^{sS_{i,j}}}, \tag{19}$$

where $S_{i,y_i} = \cos(\theta_{i,y_i}) - m$ and $S_{i,j} = \cos\theta_{i,j}$, $m$ is a margin hyper-parameter, and $s$ is a scale hyper-parameter. It shows that cosine-based softmax losses are quite similar to pair-based angular losses, as both of them optimize embeddings in an angular space. The minor difference is that cosine-based softmax losses calculate the similarities between an embedding and a class template, *i.e.*, $\cos\theta_{i,k} = \langle\hat{f}_i, \hat{w}_k\rangle$, and a margin is usually introduced.

This motivates us to figure out whether cosine-based softmax losses also suffer from the analogous unbalanced direction updating to that of pair-based angular losses. In the same way, the gradient of a

Table 1: Deep metric learning datasets.

| Name | Num. of Classes | | Num. of Samples | |
|------|-------|------|--------|--------|
| | Train | Test | Train | Test |
| CUB200-2011 | 100 | 100 | 5,864 | 5,924 |
| Cars196 | 98 | 98 | 8,054 | 8,131 |
| SOP | 11,318 | 11,316 | 59,551 | 60,502 |
| In-Shop | 3997 | 3985 | 25,882 | 26,830 |

Table 2: The effect of hyper-parameter $\eta$.

| $\eta$ | Cars196 | | | | | |
|--------|-----|----|-----|-----|-----|-----|
| | NMI | F1 | R@1 | R@2 | R@4 | R@8 |
| 0 | 56.66 | 24.44 | 60.79 | 71.30 | 79.47 | 86.27 |
| 0.1 | 59.08 | **26.50** | 66.72 | 76.92 | 84.39 | 89.88 |
| 0.5 | **59.17** | 25.51 | **67.89** | **78.56** | **85.59** | **90.99** |
| 1.0 | 58.43 | 24.61 | 64.97 | 75.92 | 84.18 | 90.03 |
| 1.5 | 56.38 | 22.37 | 57.72 | 70.42 | 80.04 | 87.43 |

cosine-based softmax loss to an embedding is computed as follows:

$$\frac{\partial L}{\partial f_i} = \sum_{(i,k)} \frac{\partial L}{\partial S_{i,k}} \frac{\partial S_{i,k}}{\partial \cos\theta_{i,k}} \frac{\partial \cos\theta_{i,k}}{\partial f_i} = \sum_{(i,k)} \phi'_{i,k} \frac{\partial \cos\theta_{i,k}}{\partial f_i} = \sum_{(i,k)} \phi'_{i,k} \frac{1}{||f_i||_2}(\hat{w}_k - \cos\theta_{i,k}\hat{f}_i), \quad (20)$$

where $\phi'_{i,k} = \frac{\partial L}{\partial \cos\theta_{i,k}}$. It has a similar structure to Equation 4 in which the magnitude of the gradient is inversely proportional to $||f_i||_2$.

On the other hand, contrastive self-supervised learning algorithms also adopt the $l_2$-normalization step for embeddings and aim to maximize the cosine distances of positive embedding pairs and minimize those of negative embedding pairs with contrastive losses [19, 20, 21], which could be regarded as variants of normalized $N$-pair loss as in Equation 3. Therefore, we consider that the above analysis of pair-based angular losses is also applicable for them, to which the proposed SEC is also beneficial.

Therefore, to reduce the influence of embedding norm on the direction updating, we further combine SEC with a cosine-based softmax (c-softmax) loss function or a contrastive loss function,

$$L = L_{\text{c-softmax}} + \eta * L_{\text{sec}}, L = L_{\text{contrastive}} + \eta * L_{\text{sec}}, \quad (21)$$

where $\eta$ is a trade-off hyper-parameter. This helps constrain the embeddings to be on the surface of the same hypersphere, and thus more balanced direction updating is performed in batch optimization.

## 4 Experiments

### 4.1 Datasets, evaluation metrics and implementation details

**(1) Deep metric learning task**: we employ four fine-grained image clustering and retrieval benchmarks, including CUB200-2011 [28], Cars196 [29], SOP [5], and In-Shop [30]. We Follow the protocol in [5, 30] to split the training and testing sets for them as in Table 1. For CUB200-2011 and Cars196, we do not use the bounding box annotations during training and testing. NMI, F1, and Recall@K are used as the evaluation metrics. The backbone network is BN-Inception [13] pretrained on ImageNet [31]. We set batch size to 120 and embedding size to 512 for all methods and datasets. We use Adam optimizer [27]. The compared methods are vanilla triplet loss ($m = 1.0$), semihard triplet loss ($m = 0.2$) [4], normalized $N$-pair loss ($s = 25$) [10, 18], and multi-similarity loss ($\epsilon = 0.1$, $\lambda = 0.5$, $\alpha = 2$, $\beta = 40$) [11], where the former two losses employ a normalized Euclidean distance and the latter two losses employ a cosine distance. **(2) Face recognition task**: CASIA-WebFace [32] is employed as the training set while the testing sets include LFW [33], AgeDB30 [34], CFP-FP [35], and MegaFace Challenge 1 [36]. We adopt ResNet50 [37] as in [26] (*i.e.*, SE-ResNet50E-IR). We set batch size to 256 and embedding size to 512 for all methods. We use SGD with momentum 0.9. The compared methods are sphereface [24], cosface [25], and arcface [26]. The hyper-parameter $s$ is set to 64 while $m$ for sphereface, cosface, and arcface are 3, 0.35, and 0.45, respectively. **(3) Contrastive self-supervised learning task**: we follow the framework and settings in SimCLR [19] and evaluate on CIFAR-10 and CIFAR-100 datasets [38]. We use ResNet-50 and a 2-layer MLP head to output 128-d embeddings, which are trained using SGD with momentum 0.9. NT-Xent with temperature 0.5 is the loss and the batch size is 256. *More details are provided in Appendix B.*

### 4.2 Ablation study and discussion

**The effect of $\eta$.** $\eta$ controls the regularization strength of SEC. We consider a triplet loss on Cars196 dataset and vary $\eta$ from 0.1 to 1.5 as in Table 2. It is seen that SEC leads to a robust improvement with $\eta \in [0.1, 1.0]$, while the result degrades with a much larger $\eta$. It indicates that SEC is not sensitive to the choice of $\eta$ with mild values and an appropriate distribution of embedding norm helps improve the generalization ability.

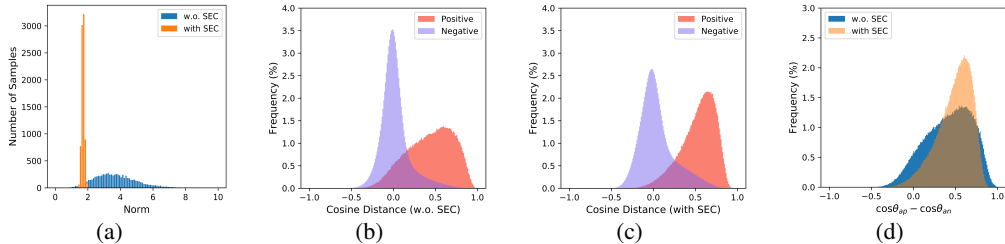

(a)                    (b)                    (c)                    (d)

Figure 5: Ablation studies of SEC with a triplet loss on Cars196 dataset. (a) The distribution of embedding norms on training set. (b) (c) The distribution of $\cos\theta_{ap}$ and $\cos\theta_{an}$ on testing set learned without and with SEC, respectively. (d) The distribution of $\cos\theta_{ap} - \cos\theta_{an}$ on testing set.

**The effect of SEC.** In this part, we employ the triplet loss to investigate the effect of SEC from different perspectives. In Figure 5(a), we observe that for triplet loss with SEC, the learned norms have a more compact distribution, showing the explicit effect of SEC. From the previous analysis in Section 3.3, we explain that more similar norms would lead to a more balanced direction update among different embeddings. To verify its effectiveness, we consider whether this update results in better embedding directions, and here we illustrate this quality by cosine distances of positive and negative pairs in testing set. From Figure 5(b) and 5(c), we observe that for positive pairs, the distribution of their cosine distances becomes more compact, while the distribution still remains compact for negative pairs. It indicates that SEC helps learn a relatively class-independent distance metric [18], which benefits the models' generalization ability. Besides, the distribution of $(\cos\theta_{ap} - \cos\theta_{an})$ is also studied in Figure 5(d). We observe that the number of the triplets violating $\cos\theta_{ap} > \cos\theta_{an}$ decreases with SEC, indicating that SEC helps learn the required embedding distribution. In summary, better performances are achieved with SEC by explicitly decreasing the norm variance while implicitly learning more discriminative embedding direction. More illustrations are provided in Appendix C.

**Convergence rate.** We analyze the convergence of deep metric learning with and without SEC in Figure 4. From the figure, we have two important observations. First, when combined with SEC, loss functions converge much faster than the original loss functions, *e.g.*, triplet loss with learning rate $1e-5$ in Figure 4(a), and also obtain a much better perfor-

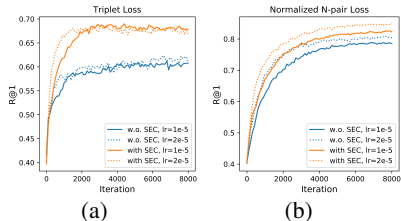

(a)                    (b)

Figure 4: Testing R@1 on Cars196 dataset learned with and without SEC.

mance. We attribute this fast convergence to the regularization power of SEC, which enforces a strong constraint on the distribution of the embedding norm and achieves a more balanced update. Second, when we increase the learning rate properly, the convergence rate of loss functions with SEC becomes faster and similar results are also received than those of the original learning rate, as shown in Figure 4(b). This observation indicates that loss functions with SEC may be less sensitive to the learning rate and a larger learning rate also leads to a faster convergence.

## 4.3 Quantitative results on three tasks

**(1) Deep metric learning**: we evaluate four methods on fine grained image retrieval and clustering tasks. In particular, the comparisons are performed between four representative baseline loss functions with and without SEC. The results are provided in Table 3. As shown in the table, semihard triplet loss performs better than triplet loss, indicating that the well-designed hard example mining strategies are effective. Normalized $N$-pair loss also achieves a better performance than the triplet loss as it allows a joint comparison among more than one negative example in the optimization. Multi-similarity loss behaves much better than the other three loss functions as it considers three different similarities for pair weighting. Further, we observe that SEC consistently boosts the performance of these four loss functions on the four datasets. On CUB200-2011, SEC shows a significant improvement on NMI, F1, and R@1 of triplet loss by 4.39%, 7.44%, and 7.48%, respectively. Specifically, for the state-of-the-art method multi-similarity loss, on Cars196 dataset, SEC also improvs the NMI, F1, and R@1 by 3.72%, 4.36%, and 1.66%, respectively. On one hand, this shows the superiority of SEC as it constrains the embeddings to be on the surface of the same hypersphere and results in more balanced direction update. On the other hand, it also demonstrates that SEC is available for a wide range of pair-based loss functions and further benefits their generalization abilities. **(2) Face recognition**: we

Table 3: Experimental results of deep metric learning. NMI, F1, and Recall@K are reported.

| Method | CUB200-2011 | | | | | | Cars196 | | | | | |
|---|---|---|---|---|---|---|---|---|---|---|---|---|
| | NMI | F1 | R@1 | R@2 | R@4 | R@8 | NMI | F1 | R@1 | R@2 | R@4 | R@8 |
| Triplet Loss | 59.85 | 23.39 | 53.34 | 65.60 | 76.30 | 84.98 | 56.66 | 24.44 | 60.79 | 71.30 | 79.47 | 86.27 |
| Triplet Loss + $L^2$-reg | 60.11 | 24.03 | 54.81 | 66.21 | 76.87 | 84.91 | 56.65 | 23.95 | 63.02 | 72.97 | 80.79 | 86.85 |
| Triplet Loss + SEC | 64.24 | 30.83 | 60.82 | 71.61 | 81.40 | 88.86 | 59.17 | 25.51 | 67.89 | 78.56 | 85.59 | 90.99 |
| Semihard Triplet [4] | 69.66 | 40.30 | 65.31 | 76.45 | 84.71 | 90.99 | 67.64 | 38.31 | 80.17 | 87.95 | 92.49 | 95.67 |
| Semihard Triplet + $L^2$-reg | 70.50 | 41.39 | 65.60 | 76.81 | 84.89 | 90.82 | 69.24 | 40.24 | 82.60 | 89.44 | 93.54 | 96.19 |
| Semihard Triplet + SEC | 71.62 | 42.05 | 67.35 | 78.73 | 86.63 | 91.90 | 72.67 | 44.67 | 85.19 | 91.53 | 95.28 | 97.29 |
| Normalized N-pair Loss | 69.58 | 40.23 | 61.36 | 74.36 | 83.81 | 89.94 | 68.07 | 37.83 | 78.59 | 87.22 | 92.88 | 95.94 |
| Normalized N-pair Loss + $L^2$-reg | 69.73 | 40.08 | 64.58 | 76.03 | 84.74 | 91.12 | 69.20 | 39.13 | 81.87 | 88.85 | 93.47 | 96.54 |
| Normalized N-pair Loss + SEC | 72.24 | 43.21 | 66.00 | 77.23 | 86.01 | 91.83 | 70.61 | 42.12 | 82.29 | 89.60 | 94.26 | 97.07 |
| Multi-Similarity [11] | 70.57 | 40.70 | 66.14 | 77.03 | 85.43 | 91.26 | 70.23 | 42.13 | 84.07 | 90.23 | 94.12 | 96.53 |
| Multi-Similarity + $L^2$-reg | 70.89 | 41.71 | 65.67 | 76.98 | 85.21 | 91.19 | 71.00 | 42.55 | 84.82 | 90.95 | 94.59 | 96.69 |
| Multi-Similarity + SEC | 72.85 | 44.82 | 68.79 | 79.42 | 87.20 | 92.49 | 73.95 | 46.49 | 85.73 | 91.96 | 95.51 | 97.54 |

| Method | SOP | | | | | | In-Shop | | | | | |
|---|---|---|---|---|---|---|---|---|---|---|---|---|
| | NMI | F1 | R@1 | R@10 | R@100 | R@1000 | R@1 | R@10 | R@20 | R@30 | R@40 | R@50 |
| Triplet Loss | 88.67 | 29.61 | 62.69 | 80.39 | 91.89 | 97.86 | 82.12 | 95.18 | 96.83 | 97.54 | 97.95 | 98.26 |
| Triplet Loss + $L^2$-reg | 88.93 | 30.91 | 64.07 | 81.27 | 92.18 | 97.93 | 83.01 | 95.46 | 96.85 | 97.45 | 97.94 | 98.28 |
| Triplet Loss + SEC | 89.68 | 34.29 | 68.86 | 83.76 | 92.93 | 98.00 | 85.29 | 96.29 | 97.48 | 97.99 | 98.34 | 98.57 |
| Semihard Triplet [4] | 91.16 | 41.89 | 74.46 | 88.16 | 95.21 | 98.59 | 87.16 | 97.11 | 98.17 | 98.54 | 98.76 | 98.98 |
| Semihard Triplet + $L^2$-reg | 91.16 | 41.77 | 74.88 | 88.25 | 95.18 | 98.53 | 88.04 | 97.39 | 98.24 | 98.65 | 98.83 | 98.99 |
| Semihard Triplet + SEC | 91.72 | 44.90 | 77.59 | 90.12 | 96.04 | 98.80 | 89.68 | 97.95 | 98.61 | 98.94 | 99.09 | 99.21 |
| Normalized N-pair Loss | 90.97 | 41.21 | 74.30 | 87.81 | 95.12 | 98.55 | 86.43 | 96.99 | 98.00 | 98.40 | 98.70 | 98.93 |
| Normalized N-pair Loss + $L^2$-reg | 91.12 | 41.73 | 75.11 | 88.42 | 95.15 | 98.53 | 86.54 | 96.98 | 98.06 | 98.52 | 98.73 | 98.85 |
| Normalized N-pair Loss + SEC | 91.49 | 43.75 | 76.89 | 89.64 | 95.77 | 98.68 | 88.63 | 97.60 | 98.45 | 98.77 | 99.01 | 99.14 |
| Multi-Similarity [11] | 91.42 | 43.33 | 76.29 | 89.38 | 95.58 | 98.58 | 88.11 | 97.55 | 98.34 | 98.76 | 98.94 | 99.09 |
| Multi-Similarity + $L^2$-reg | 91.65 | 44.51 | 77.34 | 89.61 | 95.67 | 98.65 | 88.51 | 97.59 | 98.50 | 98.84 | 99.03 | 99.12 |
| Multi-Similarity + SEC | 91.89 | 46.04 | 78.67 | 90.77 | 96.15 | 98.76 | 89.87 | 97.94 | 98.80 | 99.06 | 99.24 | 99.35 |

Table 4: Experimental results of face recognition. Face verification accuracy is reported on LFW, AgeDB30, and CFPFP while face identification accuracy is reported on MegaFace.

| Method | Face Verification | | | Size of MegaFace Distractors | | | |
|---|---|---|---|---|---|---|---|
| | LFW | AgeDB30 | CFPFP | $10^6$ | $10^5$ | $10^4$ | $10^3$ |
| Softmax | 98.97 | 91.30 | 93.39 | 80.43 | 87.11 | 92.83 | 96.12 |
| Sphereface [24] | 99.20 | 93.45 | 94.24 | 87.72 | 92.48 | 95.64 | 97.68 |
| Sphereface + $L^2$-reg | 99.28 | 93.42 | 94.30 | 88.38 | 92.86 | 95.93 | 97.87 |
| Sphereface + SEC | 99.30 | 93.45 | 94.39 | 88.42 | 92.79 | 95.88 | 97.86 |
| Cosface [25] | 99.37 | 93.82 | 94.46 | 90.71 | 94.30 | 96.57 | 98.09 |
| Cosface + $L^2$-reg | 99.12 | 94.32 | 94.64 | 91.03 | 94.46 | 96.85 | 98.24 |
| Cosface + SEC | 99.42 | 94.37 | 94.93 | 91.13 | 94.63 | 96.92 | 98.37 |
| Arcface [26] | 99.22 | 94.18 | 94.69 | 90.31 | 94.07 | 96.67 | 98.20 |
| Arcface + $L^2$-reg | 99.32 | 93.93 | 94.77 | 90.68 | 94.34 | 96.83 | 98.32 |
| Arcface + SEC | 99.35 | 93.82 | 94.91 | 90.91 | 94.56 | 96.95 | 98.37 |

Table 5: Experimental results of contrastive self-supervised learning with SimCLR [19]. Top 1/5 accuracy of linear evaluation is reported.

| Method | Training | CIFAR-10 | | CIFAR-100 | |
|---|---|---|---|---|---|
| | Epoch | Top 1 | Top 5 | Top 1 | Top 5 |
| NT-Xent [19] | 100 | 84.76 | 99.36 | 58.43 | 85.26 |
| NT-Xent + $L^2$-reg | | 86.64 | 99.56 | 61.43 | 87.23 |
| NT-Xent + SEC | | 86.87 | 99.64 | 61.66 | 87.33 |
| NT-Xent [19] | 200 | 89.05 | 99.69 | 65.73 | 89.64 |
| NT-Xent + $L^2$-reg | | 90.14 | 99.73 | 66.57 | 90.18 |
| NT-Xent + SEC | | 90.35 | 99.77 | 66.25 | 90.12 |

consider three cosine-based softmax loss functions with and without SEC. The results are provided in Table 4. As shown in Table 4, three cosine-based softmax loss functions perform better than the original softmax by adopting angular margin for a better discriminative ability. We also observe that SEC improves the performance of sphereface, cosface, and arcface in most cases. For example, on MegaFace dataset with $10^6$ distractors, the rank-1 accuracies of sphereface, cosface, and arcface are boosted by 0.7%, 0.42%, and 0.6%, respectively. It illustrates that these cosine-based softmax losses also benefit from SEC, helping learn a more discriminative embedding space. **(3) Contrastive self-supervised learning**: we consider the latest SimCLR framework using NT-Xent loss with and without SEC. The results are provided in Table 5. From the table, we observe that SimCLR benefits from more training steps, while SEC also consistently enhances its performance on the two datasets with two different settings of training epochs. For instance, on CIFAR-10 dataset, SEC improves the top-1 linear evaluation accuracy of SimCLR by 2.11% and 1.3% when training for 100 and 200 epochs, respectively. It shows that SEC is also helpful for contrastive self-supervised learning methods to learn more useful visual representations.

In the above tables, the results of $L^2$-reg is also illustrated, for which the weight $\eta$ is carefully tuned to obtain the best performance. We observe that SEC consistently outperforms $L^2$-reg on deep metric learning, while SEC obtains slightly better results than $L^2$-reg in most cases on face recognition and contrastive self-supervised learning, indicating the superiority of SEC on various tasks.

## 5 Conclusion

In this paper, we investigate the problem of deep metric learning with spherical embedding constraint. In particular, we first investigate the importance of the embedding norm distribution for deep metric learning with angular distance, and then propose a spherical embedding constraint (SEC) to reduce the variance of the embedding norm distribution. SEC adaptively pushes the embeddings to be on the same hypersphere and achieves a more balanced direction update. Extensive experiments on deep metric learning, face recognition, and contrastive self-supervised learning show that the SEC-based angular space learning strategy helps improve the generalization performance of the state-of-the-art.

## Broader Impact

In this paper, we mainly investigate the effect of embedding norm to the direction update in the existing angular loss functions and how to improve the angular distance optimization. Our experiments indicate that the proposed SEC would be beneficial for applications related to discriminative representation learning of images in an angular space, where experiments on face recognition are also conducted. However, we note that although face recognition is quite controversial as a technique, there is no reason to expect that the mild improvement brought by SEC to the face recognition performance should make substantial difference to its societal application, nor is it expected to exacerbate its e.g. racial unbalances. As for the existing society and ethical problems of face recognition, we also agree that further study is still needed to make a substantial improvement before it is widely used in real life.

## Acknowledgement

This work is supported in part by Science and Technology Innovation 2030 –"New Generation Artificial Intelligence" Major Project (No. 2018AAA0100904), National Key R&D Program of China (No. 2018YFB1403600), NSFC (No. 61672456, 61702448, U19B2043), Artificial Intelligence Research Foundation of Baidu Inc., the funding from HIKVision and Horizon Robotics, and ZJU Converging Media Computing Lab.

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
