[Supplementary Material]

# Supplementary Material for "Deep Metric Learning with Spherical Embedding"

**Dingyi Zhang[1], Yingming Li[1]\*, Zhongfei Zhang[2]**
[1]College of Information Science & Electronic Engineering, Zhejiang University, China
[2]Department of Computer Science, Binghamton University, USA
{dyzhang, yingming}@zju.edu.cn, zhongfei_mark@yahoo.com

## A   The proof of Proposition 4 and 5

### A.1

When adopting the embedding normalization for angular distance calculation, we show that the gradient of a pair-based loss function $L$ to the embedding $f$ is:

$$\frac{\partial L}{\partial f} = (\frac{\partial \hat{f}}{\partial f})^\top \frac{\partial L}{\partial \hat{f}} = \frac{1}{||f||_2}(I - \hat{f}\hat{f}^\top)\frac{\partial L}{\partial \hat{f}}, \tag{1}$$

where $I - \hat{f}\hat{f}^\top$ projects the gradient to the tangent hyperplane of $f$. The SGD with momentum method would update the embedding by

$$v_{t+1} = \beta v_t + \frac{\partial L}{\partial f_t} \tag{2}$$

$$f_{t+1} = f_t - \alpha v_{t+1}. \tag{3}$$

**Proposition 4.** *When using SGD with momentum, the embedding direction is updated by*

$$\hat{f}_{t+1} = \hat{f}_t - \frac{\alpha}{||f_t||_2^2}(I - \hat{f}_t\hat{f}_t^\top)[||f_t||_2\beta v_t + (I - \hat{f}_t\hat{f}_t^\top)\frac{\partial L}{\partial \hat{f}_t}] + O(\alpha^2). \tag{4}$$

*Proof.* Based on Equation 3, we have

$$||f_{t+1}||_2^2 = ||f_t||_2^2 - 2\alpha f_t^\top v_{t+1} + \alpha^2 v_{t+1}^\top v_{t+1},$$

and thus

$$||f_{t+1}||_2 = \sqrt{||f_t||_2^2[1 - \frac{2\alpha}{||f_t||_2}\hat{f}_t^\top v_{t+1} + \frac{\alpha^2}{||f_t||_2^2}v_{t+1}^\top v_{t+1}]} = ||f_t||_2 - \alpha\hat{f}_t^\top v_{t+1} + O(\alpha^2).$$

From Equation 3, we also have

$$||f_{t+1}||_2\hat{f}_{t+1} = ||f_t||_2\hat{f}_t - \alpha v_{t+1},$$

Then we combine the above results with Equation 1 and 2 and we have

$$\begin{aligned}
\hat{f}_{t+1} &= \frac{||f_t||_2}{||f_{t+1}||_2}\hat{f}_t - \frac{\alpha}{||f_{t+1}||_2}v_{t+1} \\
&= (1 + \frac{\alpha}{||f_t||_2}\hat{f}_t^\top v_{t+1})\hat{f}_t - \frac{\alpha}{||f_t||_2}v_{t+1} + O(\alpha^2) \\
&= \hat{f}_t - \frac{\alpha}{||f_t||_2}(I - \hat{f}_t\hat{f}_t^\top)v_{t+1} + O(\alpha^2) \\
&= \hat{f}_t - \frac{\alpha}{||f_t||_2}(I - \hat{f}_t\hat{f}_t^\top)[\beta v_t + \frac{1}{||f_t||_2}(I - \hat{f}_t\hat{f}_t^\top)\frac{\partial L}{\partial \hat{f}_t}] + O(\alpha^2) \\
&= \hat{f}_t - \frac{\alpha}{||f_t||_2^2}(I - \hat{f}_t\hat{f}_t^\top)[||f_t||_2\beta v_t + (I - \hat{f}_t\hat{f}_t^\top)\frac{\partial L}{\partial \hat{f}_t}] + O(\alpha^2).
\end{aligned}$$

---

## A.2

Adam would update the embedding by

$$v_{t+1} = \beta_1 v_t + (1-\beta_1)\frac{\partial L}{\partial f_t}, \ g_{t+1} = \beta_2 g_t + (1-\beta_2)\|\frac{\partial L}{\partial f_t}\|_2^2 \tag{5}$$

$$f_{t+1} = f_t - \alpha\frac{v_{t+1}/(1-\beta_1^t)}{\sqrt{g_{t+1}/(1-\beta_2^t)}+\epsilon}. \tag{6}$$

**Proposition 5.** *With Adam, the embedding direction is updated by*

$$\hat{f}_{t+1} = \hat{f}_t - \frac{\alpha}{\|f_t\|_2}(I - \hat{f}_t\hat{f}_t^\top)\frac{\sqrt{1-\beta_2^t}[\|f_t\|_2\beta_1 v_t + (1-\beta_1)(I - \hat{f}_t\hat{f}_t^\top)\frac{\partial L}{\partial \hat{f}_t}]}{(1-\beta_1^t)\sqrt{\|f_t\|_2^2\beta_2 g_t + (1-\beta_2)(\frac{\partial L}{\partial \hat{f}_t})^\top(I - \hat{f}_t\hat{f}_t^\top)\frac{\partial L}{\partial \hat{f}_t}}} + O(\alpha^2). \tag{7}$$

*Proof.* Based on Equation 6, we have

$$\|f_{t+1}\|_2^2 = \|f_t\|_2^2 - 2\alpha\frac{\sqrt{1-\beta_2^t}f_t^\top v_{t+1}}{(1-\beta_1^t)\sqrt{g_{t+1}}} + \alpha^2[\frac{\sqrt{1-\beta_2^t}}{(1-\beta_1^t)\sqrt{g_{t+1}}}]^2 v_{t+1}^\top v_{t+1},$$

where we neglect $\epsilon$ for simplicity. Therefore,

$$\|f_{t+1}\|_2 = \sqrt{\|f_t\|_2^2\{1 - 2\alpha\frac{\sqrt{1-\beta_2^t}\hat{f}_t^\top v_{t+1}}{\|f_t\|_2(1-\beta_1^t)\sqrt{g_{t+1}}} + \frac{\alpha^2}{\|f_t\|_2^2}[\frac{\sqrt{1-\beta_2^t}}{(1-\beta_1^t)\sqrt{g_{t+1}}}]^2 v_{t+1}^\top v_{t+1}\}}$$

$$= \|f_t\|_2 - \alpha\frac{\sqrt{1-\beta_2^t}\hat{f}_t^\top v_{t+1}}{(1-\beta_1^t)\sqrt{g_{t+1}}} + O(\alpha^2).$$

From Equation 6 we also have

$$\|f_{t+1}\|_2\hat{f}_{t+1} = \|f_t\|_2\hat{f}_t - \alpha\frac{v_{t+1}/(1-\beta_1^t)}{\sqrt{g_{t+1}/(1-\beta_2^t)}}.$$

Then combining the above results with Equation 1 and 5, we obtain

$$\hat{f}_{t+1} = \frac{\|f_t\|_2}{\|f_{t+1}\|_2}\hat{f}_t - \frac{\alpha v_{t+1}/(1-\beta_1^t)}{\|f_{t+1}\|_2\sqrt{g_{t+1}/(1-\beta_2^t)}}$$

$$= [1 + \frac{\alpha\sqrt{1-\beta_2^t}\hat{f}_t^\top v_{t+1}}{\|f_t\|_2(1-\beta_1^t)\sqrt{g_{t+1}}}]\hat{f}_t - \frac{\alpha\sqrt{1-\beta_2^t}v_{t+1}}{\|f_t\|_2(1-\beta_1^t)\sqrt{g_{t+1}}} + O(\alpha^2)$$

$$= \hat{f}_t - \frac{\alpha}{\|f_t\|_2}(I - \hat{f}_t\hat{f}_t^\top)\frac{\sqrt{1-\beta_2^t}v_{t+1}}{(1-\beta_1^t)\sqrt{g_{t+1}}} + O(\alpha^2)$$

$$= \hat{f}_t - \frac{\alpha}{\|f_t\|_2}(I - \hat{f}_t\hat{f}_t^\top)\frac{\sqrt{1-\beta_2^t}[\|f_t\|_2\beta_1 v_t + (1-\beta_1)(I - \hat{f}_t\hat{f}_t^\top)\frac{\partial L}{\partial \hat{f}_t}]}{(1-\beta_1^t)\sqrt{\|f_t\|_2^2\beta_2 g_t + (1-\beta_2)(\frac{\partial L}{\partial \hat{f}_t})^\top(I - \hat{f}_t\hat{f}_t^\top)\frac{\partial L}{\partial \hat{f}_t}}} + O(\alpha^2).$$

# B  More implementation details

## B.1  Deep metric learning

During training, we follow [1] and adopt random resized cropping for data augmentation. Specifically, each image is first resized so that the length of its shorter side is 256. Then a random crop is generated with scale varying in $[0.16, 1.0]$ and aspect ratio varying in $[\frac{3}{4}, \frac{4}{3}]$. Finally, this crop is resized to 227 by 227 and randomly horizontally flipped. During testing, after the image is resized to have a shorter side with length 256, it is only center cropped to 227 by 227. The parameters of batch normalization layers are frozen during training. To construct a mini-batch, we first randomly sample $C$ different classes and than randomly sample $K$ images from each class. For triplet loss, semihard triplet loss, normalized $N$-pair loss, and multi-similarity loss, the values of $K$ are 3, 3, 2, and 5, respectively. On top of the final average pooling layer of the backbone network, we add a head to output 512-d embeddings. This head is composed of a BN layer and a FC layer for triplet loss, semihard triplet loss, and normalized $N$-pair loss. For multi-similarity loss, the head is only a FC layer when we do not use SEC, and the head composes of a BN layer and a FC layer when using SEC. We experimentally find that such head settings bring better results. Other training settings are listed in Table 1.

Table 1: Hyper-parameters for deep metric learning task. We use T, SHT, NNP, and MS to denote triplet loss, semihard triplet loss, normalized $N$-pair loss, and multi-similarity loss, respectively.

| Dataset | Iters | Loss | LR Settings (lr for head/lr for backbone/lr decay@iter) |
|---|---|---|---|
| CUB200-2011 | 8k | T, SHT | 0.5e-5/2.5e-6/0.1@5k |
| | | NNP | 1e-5/5e-6/0.1@5k |
| | | MS | 5e-5/2.5e-5/0.1@3k, 6k |
| Cars196 | 8k | T, SHT | 1e-5/1e-5/0.5@4k, 6k |
| | | NNP [*] | 1e-5/1e-5/0.5@4k, 6k |
| | | MS | 4e-5/4e-5/0.1@2k |
| SOP | 12k | T, SHT, NNP, MS | 5e-4/1e-4/0.1@6k |
| In-Shop | 12k | T, SHT, NNP, MS | 5e-4/1e-4/0.1@6k |

[*] For NNT on Cars196 dataset, the lr settings of 2e-5/2e-5/0.5@4k, 6k would bring a better result.

Table 2: The settings of $\eta$ for SEC and $L^2$-reg in Table 3 of the original paper. We use T, SHT, NNP, and MS to denote triplet loss, semihard triplet loss, normalized $N$-pair loss, and multi-similarity loss, respectively.

| Dataset | $\eta$ for SEC/$L^2$-reg | | | |
|---|---|---|---|---|
| | T | SHT | NNP | MS |
| CUB | 1.0 / 1e-4 | 0.5 / 1e-3 | 1.0 / 1e-2 | 0.5 / 5e-3 |
| Cars | 0.5 / 1e-4 | 0.5 / 1e-2 | 1.0 / 1e-2 | 1.0 / 1e-2 |
| SOP | 1.0 / 1e-4 | 1.0 / 5e-4 | 1.0 / 1e-3 | 0.5 / 5e-4 |
| In-Shop | 1.0 / 5e-5 | 1.0 / 5e-5 | 1.0 / 1e-3 | 0.25 / 1e-4 |

The hyper-parameters of compared losses are: (1) $m = 1.0$ for vanilla triplet loss, following [2]. (2) $m = 0.2$ for semihard triplet loss, following [3]. (3) $s = 25$ for normalized $N$-pair loss, where we test two settings: $s = 25$ and $s = 64$, and we find that the former is better. (4) $\epsilon = 0.1$, $\lambda = 0.5$, $\alpha = 2$, and $\beta = 40$ for multi-similarity loss, following the original authors' GitHub (which is a little different from the original paper). When combined with different loss functions, we tune $\eta$ in $\{0.25, 0.5, 1.0\}$ for SEC and in $\{1e-2, 5e-3, 1e-3, 5e-4, 1e-4, 5e-5, 1e-5\}$ for $L^2$-reg. The detailed settings of them are listed in Table 2. The model is trained on a NVIDIA 2080Ti GPU.

## B.2  Face recognition

We train the model for 16 epochs with the learning rate starting from 0.1 and divided by 10 at 10, 14 epochs. For the hyper-parameters of the three compared loss functions, they largely follow the original papers, while the model does not converge on CASIA-WebFace with the value of $m$ in the original papers for sphereface ($m = 4$) and arcface ($m = 0.5$), we thus choose the slightly smaller values. In Table 4 of the original paper, when combined with SEC or $L^2$-reg, $\eta$ is set to 0 during the first three epochs, linearly increasing at the 4th epoch, and unchanged for the following epochs. When combined with different loss functions, we tune $\eta$ in $\{0.5, 1.0, 5.0, 10.0\}$ for SEC and in $\{5e-2, 1e-2, 5e-3, 1e-3, 5e-4\}$ for $L^2$-reg. The detailed settings of them are provided in Table 6.

## B.3  Contrastive self-supervised learning

The ResNet-50 backbone network and the 2-layer MLP head are trained with cosine decayed learning rate starting from 0.5, where the learning rate is tuned in $\{0.5, 1.0, 1.5\}$. For linear evaluation, a linear classifier is trained for 100 epochs using SGD with momentum 0.9 with batch size 256 and the learning rate starts from 5 and is divided by 5 at 60, 75, 90 epochs, where the learning rate is tuned in $\{1, 2, 5, 10\}$. When combined with SEC or $L^2$-reg, we use a linearly increasing $\eta$ during the whole training stage, *i.e.*,

$$\eta_t = \eta * \frac{t}{\text{num. of total iterations}}, \tag{8}$$

at the $t$-th iteration. In Table 5 of the original paper, we tune $\eta$ in $\{0.01, 0.05, 0.1, 0.25, 0.5, 1.0\}$ for SEC and in $\{5e-2, 1e-2, 5e-3, 1e-3, 5e-4\}$ for $L^2$-reg. The detailed settings of them are provided in Table 7. We use global BN as in [4]. The data augmentation includes random flip, random crop and resize, and color distortions.

Figure 1: Comparisons between SEC and $L^2$-reg with the triplet loss on Cars196 dataset. We show the R@1 on testing set, the average norm, and the variance of norm in a mini-batch during training.

Table 3: The effect of $\eta$ when using $L^2$-reg with the triplet loss on Cars196 dataset.

| $\eta$ | Cars196 | | |
|---|---|---|---|
| | NMI | F1 | R@1 |
| baseline (w.o. $L^2$-reg) | 56.49 | 23.72 | 60.84 |
| 1e-3 | 55.95 | 22.20 | 57.16 |
| 5e-4 | 56.20 | 23.16 | 61.35 |
| 1e-4 | **56.65** | 23.95 | **63.02** |
| 5e-5 | **56.65** | **24.32** | 61.48 |

## C  More explanations and illustrations

### C.1  Discussion and comparison between SEC and $L^2$-reg

In addition to SEC, here we also discuss another norm regularization method which is empirically helpful for angular pair-based losses. This method is proposed in [5] to regularize the $l_2$-norm of embeddings to be small, which we refer as $L^2$-reg, *i.e.*,

$$L_{L^2\text{-reg}} = \frac{1}{N} \sum_{i=1}^{N} ||f_i||_2^2. \qquad (9)$$

We note that $L^2$-reg could be seen as a special case of SEC by setting $\mu = 0$. During training, the complete objective function is $L = L_{\text{metric}} + \eta * L_{L^2\text{-reg}}$. We study the effect of this method with the triplet loss on Cars196 dataset as in Figure 1. Here $\eta$ is carefully tuned to obtain the best performance for $L^2$-reg, which will be compared with the original SEC and a variant of SEC with fixed $\mu$ (we fix $\mu$ to the initial average norm of the training set, $8.70$).

We first notice in Figure 1 that, as expected, $L^2$-reg would decrease the average norm, while with the norm becoming small, it also has the side effect of reducing the norm variance, which is similar to the function of SEC. Previous works have not stated that which factor has more impacts on the performance, while here we argue that reducing the variance is more important than reducing the norm, since in Figure 1, an improved result compared with $L^2$-reg is obtained when we fix the average norm and reduce the variance more strongly by using SEC with fixed $\mu$. Therefore, we speculate that the effectiveness of $L^2$-reg also comes from the reduction of the variance. From Figure 1, we also conclude that the way SEC adjusts the norm distribution is better than $L^2$-reg, since it obtains clearly favorable results, although both of them would alter the mean and variance of the norm. Besides, SEC is more preferable in its convenience to determine $\eta$, to which the result is not very sensitive as in Table 2 of the original paper compared to the situation of $L^2$-reg as in Table 3 here.

### C.2  An empirical illustration of more balanced direction update provided by SEC

In this part, we adopt the triplet loss trained on Cars196 dataset with and without SEC to show the effect of SEC to perform more balanced direction update for embeddings. One characteristic of triplet loss is its simple formulation of gradients, *i.e.*, $|\frac{\partial L}{\partial S_{ij}}|$ is always 0 or 1 no matter which pair $(i, j)$ is considered, and $\frac{\partial S_{ij}}{\partial \hat{f}_i} = 2(\hat{f}_i - \hat{f}_j)$. Therefore, from Proposition 5, we find that the direction update of an embedding $f$ would largely rely on $\frac{\alpha}{||f||_2}$. Due to this reason, we choose triplet loss and illustrate the direction variation of each sample in the training set per 1000 iterations. The results are shown in Figure 2, where we also calculate the variance of the direction variation distribution and provide them in the legend of each sub-figure. We observe that when training with SEC, the distribution of direction variation of different embeddings clearly becomes more compact than the distribution without SEC, such as 1k $\to$ 2k, 2k $\to$ 3k, 3k $\to$ 4k, and 4k $\to$ 5k, while the two distributions are similar in other situations. It indicates that different embeddings obtain more balanced direction updates with the help of SEC, which explicitly constrains embeddings to lie on the surface of the same hypersphere. Consequently, different embeddings would all attain adequate attention from the model during training, which also benefits the generalization ability of the model.

Figure 2: An empirical illustration for explaining the effect of SEC to perform more balanced direction update. We illustrate the distribution of direction variation of all embeddings in training set per 1000 iterations. The variance of the direction variation distribution is also calculated and provided in the legend of each sub-figure.

Figure 3: Visualization of six channels of the last feature maps which have the maximal average activations. Here we use the multi-similarity loss as the baseline loss.

## C.3 Visualization of feature maps and retrieval results

In Figure 3, we further visualize the the feature maps from the last convolutional layer learned by multi-similarity loss without and with SEC. Here we only choose six channels with the maximal average activation. We observe that in some of these channels, the baseline loss may focus on not only the target object but also unrelated part or background, while with SEC, the model concentrates more on the object itself. Therefore, SEC would also benefit the model by refining the feature maps to better attend to the object region. Besides, we also illustrate some retrieval results in Figure 4. We notice that no matter the top-1 retrieval result is correct or incorrect, SEC clearly finds images which

Figure 4: Top 3 retrieved images without and with SEC. Here we use the multi-similarity loss as the baseline loss. Correct results are highlighted with *green*, while incorrect *red*.

Table 4: The effect of $\rho$ when employing a triplet loss with SEC (EMA) on Cars196 dataset. The initial average norm of all embeddings in the training set is 8.70.

| $\rho$ | Cars196 | | | Final Norm | |
|---|---|---|---|---|---|
| | NMI | F1 | R@1 | Mean | Var. |
| baseline (w.o. SEC) | 56.49 | 23.72 | 60.84 | 8.03 | 5.54 |
| 0.01 | **61.25** | **28.87** | **74.57** | 6.03 | 0.04 |
| 0.1 | 59.96 | 26.22 | 72.00 | 2.60 | 0.02 |
| 0.5 | 59.61 | 27.01 | 68.54 | 1.72 | 0.02 |
| 0.9 | 59.74 | 26.11 | 68.28 | 1.56 | 0.02 |
| 1.0* | 59.17 | 25.51 | 67.89 | 1.58 | 0.02 |

* the original version of SEC

are more similar to the query images, in terms of appearance and even the pose of the birds and cars. This observation implies that SEC would help learn a better structured embedding space with similar images closer to each other by constraining embeddings to be on a hypersphere.

# D   Spherical embedding with exponential moving average (EMA) norm

## D.1   Formulation

In this part, we further extend SEC by adopting the EMA method for updating $\mu$. With the EMA method, we aim to capture the variation of the global average norm during the training process. Specifically, we update $\mu$ at the $t$-th iteration by:

$$\mu_t = (1 - \rho)\mu_{t-1} + \rho * \frac{1}{N} \sum_{i=1}^{N} ||f_{t,i}||_2, \tag{10}$$

where $\mu_0 = \frac{1}{N} \sum_{i=1}^{N} ||f_{0,i}||_2$, $N$ is the batch size and $\rho$ is the momentum hyper-parameter in $[0, 1]$. It also helps maintain a smoothly changed $\mu$ and makes the training more stable when the average norm across mini-batches differs a lot. We note that if $\rho = 1$, this formulation degenerates to the original version of SEC. We show the influence of $\rho$ in Table 4, where the final average norm is closer to the initial one with a smaller $\rho$. From the table, we also observe consistent improvements compared with the baseline when $\rho$ is set to a proper range of values, and here the range seems to be $[0, 1]$ for the triplet loss, indicating the convenience of determining $\rho$. Besides, when employing this EMA method to update $\mu$, here we also notice a higher improvement than setting $\rho = 1.0$ (the original SEC), which indicates that this new version of SEC may be a better choice in some circumstances. A

Table 5: Experimental results of deep metric learning. NMI, F1, and Recall@K are reported.

| Method | CUB200-2011 | | | | | | Cars196 | | | | | |
|---|---|---|---|---|---|---|---|---|---|---|---|---|
| | NMI | F1 | R@1 | R@2 | R@4 | R@8 | NMI | F1 | R@1 | R@2 | R@4 | R@8 |
| Triplet Loss | 59.85 | 23.39 | 53.34 | 65.60 | 76.30 | 84.98 | 56.66 | 24.44 | 60.79 | 71.30 | 79.47 | 86.27 |
| Triplet Loss + $L^2$-reg ($\eta$=1e-4/1e-4) | 60.11 | 24.03 | 54.81 | 66.21 | 76.87 | 84.91 | 56.65 | 23.95 | 63.02 | 72.97 | 80.79 | 86.85 |
| Triplet Loss + SEC ($\eta$=1.0/0.5) | 64.24 | 30.83 | **60.82** | 71.61 | 81.40 | 88.86 | 59.17 | 25.51 | 67.89 | 78.56 | 85.59 | 90.99 |
| Triplet Loss + SEC (EMA, $\rho$=0.01, $\eta$=1.0/0.5) | **64.81** | **32.14** | 60.72 | **72.45** | **82.33** | **89.18** | **61.25** | **28.87** | **74.57** | **83.96** | **89.79** | **93.78** |
| Semihard Triplet [3] | 69.66 | 40.30 | 65.31 | 76.45 | 84.71 | 90.99 | 67.64 | 38.31 | 80.17 | 87.95 | 92.49 | 95.67 |
| Semihard Triplet + $L^2$-reg ($\eta$=1e-3/1e-2) | 70.50 | 41.39 | 65.60 | 76.81 | 84.89 | 90.82 | 69.24 | 40.24 | 82.60 | 89.44 | 93.54 | 96.19 |
| Semihard Triplet + SEC ($\eta$=0.5/0.5) | 71.62 | 42.05 | 67.35 | **78.73** | **86.63** | 91.90 | **72.67** | **44.67** | **85.19** | **91.53** | **95.28** | **97.29** |
| Semihard Triplet + SEC (EMA, $\rho$=0.01, $\eta$=0.5/0.5) | **72.00** | **43.68** | **67.51** | 77.90 | 86.44 | **91.98** | 72.38 | 44.31 | 84.73 | 91.18 | 95.07 | 97.27 |
| Normalized N-pair Loss | 69.58 | 40.23 | 61.36 | 74.36 | 83.81 | 89.94 | 68.07 | 37.83 | 78.59 | 87.22 | 92.88 | 95.94 |
| Normalized N-pair Loss + $L^2$-reg ($\eta$=1e-2/1e-2) | 69.73 | 40.08 | 64.58 | 76.03 | 84.74 | 91.12 | 69.20 | 39.13 | 81.87 | 88.85 | 93.47 | 96.54 |
| Normalized N-pair Loss + SEC ($\eta$=1.0/1.0) | **72.24** | **43.21** | **66.00** | 77.23 | 86.01 | 91.83 | 70.61 | 42.12 | 82.29 | **89.60** | **94.26** | **97.07** |
| Normalized N-pair Loss + SEC (EMA, $\rho$=0.01, $\eta$=1.0/1.0) | 71.62 | 42.16 | 65.82 | **77.31** | **86.07** | **91.98** | **70.97** | **42.60** | 81.85 | 89.30 | 93.83 | 96.57 |
| Multi-Similarity [1] | 70.57 | 40.70 | 66.14 | 77.03 | 85.43 | 91.26 | 70.23 | 42.13 | 84.07 | 90.23 | 94.12 | 96.53 |
| Multi-Similarity + $L^2$-reg ($\eta$=5e-3/1e-2) | 70.89 | 41.71 | 65.67 | 76.85 | 85.21 | 91.19 | 70.04 | 42.55 | 84.82 | 90.95 | 94.59 | 96.69 |
| Multi-Similarity + SEC ($\eta$=0.5/1.0) | 72.85 | 44.82 | 68.79 | 79.42 | 87.20 | 92.49 | **73.95** | **46.49** | **85.73** | **91.96** | **95.51** | **97.54** |
| Multi-Similarity + SEC (EMA, $\rho$=0.01, $\eta$=0.5/1.0) | **74.22** | **47.42** | **69.78** | **80.40** | **88.00** | **93.23** | 71.70 | 42.84 | 83.80 | 90.96 | 94.99 | 97.47 |

| Method | SOP | | | | | | In-Shop | | | | | |
|---|---|---|---|---|---|---|---|---|---|---|---|---|
| | NMI | F1 | R@1 | R@10 | R@100 | R@1000 | R@1 | R@10 | R@20 | R@30 | R@40 | R@50 |
| Triplet Loss | 88.67 | 29.61 | 62.69 | 80.39 | 91.89 | 97.86 | 82.12 | 95.18 | 96.83 | 97.54 | 97.95 | 98.26 |
| Triplet Loss + $L^2$-reg ($\eta$=1e-4/5e-5) | 88.93 | 30.91 | 64.07 | 81.27 | 92.18 | 97.93 | 83.01 | 95.46 | 96.85 | 97.45 | 97.94 | 98.28 |
| Triplet Loss + SEC ($\eta$=1.0/1.0) | **89.68** | **34.29** | **68.86** | **83.76** | **92.93** | **98.00** | 83.79 | 95.58 | 96.91 | 97.32 | 97.84 | **98.57** |
| Triplet Loss + SEC (EMA, $\rho$=0.01, $\eta$=1.0/1.0) | 88.64 | 29.37 | 64.03 | 79.95 | 90.71 | 97.17 | 80.38 | 94.16 | 96.12 | 96.91 | 97.47 | 97.81 |
| Semihard Triplet [3] | 91.16 | 41.89 | 74.46 | 88.16 | 95.21 | 98.59 | 87.16 | 97.11 | 98.17 | 98.54 | 98.76 | 98.98 |
| Semihard Triplet + $L^2$-reg ($\eta$=5e-4/5e-5) | 91.16 | 41.77 | 74.88 | 88.25 | 95.18 | 98.53 | 88.04 | 97.39 | 98.24 | 98.65 | 98.83 | 98.99 |
| Semihard Triplet + SEC ($\eta$=1.0/1.0) | **91.72** | **44.90** | **77.59** | **90.12** | **96.04** | **98.80** | 89.68 | **97.95** | **98.61** | **98.94** | **99.09** | **99.21** |
| Semihard Triplet + SEC (EMA, $\rho$=0.01, $\eta$=1.0/1.0) | 91.68 | 44.69 | 77.45 | 89.62 | 95.70 | 98.67 | **89.79** | 97.94 | 98.59 | 98.86 | 99.08 | 99.20 |
| Normalized N-pair Loss | 90.97 | 41.21 | 74.30 | 87.81 | 95.12 | 98.55 | 86.43 | 96.99 | 98.00 | 98.40 | 98.70 | 98.93 |
| Normalized N-pair Loss + $L^2$-reg ($\eta$=1e-3/1e-3) | 91.12 | 41.73 | 75.11 | 88.42 | 95.15 | 98.53 | 86.54 | 96.98 | 98.06 | 98.52 | 98.73 | 98.85 |
| Normalized N-pair Loss + SEC ($\eta$=1.0/1.0) | 91.49 | 43.75 | **76.89** | **89.64** | **95.77** | 98.68 | 88.63 | 97.60 | 98.45 | 98.77 | **99.01** | **99.14** |
| Normalized N-pair Loss + SEC (EMA, $\rho$=0.01, $\eta$=1.0/1.0) | **91.55** | **43.84** | 76.68 | 89.46 | 95.68 | **98.70** | **89.06** | **97.65** | **98.46** | **98.82** | 98.97 | 99.06 |
| Multi-Similarity [1] | 91.42 | 43.33 | 76.29 | 89.38 | 95.58 | 98.58 | 88.11 | 97.55 | 98.34 | 98.76 | 98.94 | 99.09 |
| Multi-Similarity + $L^2$-reg ($\eta$=5e-4/1e-4) | 91.65 | 44.51 | 77.34 | 89.61 | 95.67 | 98.65 | 88.51 | 97.59 | 98.50 | 98.84 | 99.03 | 99.12 |
| Multi-Similarity + SEC ($\eta$=0.5/0.25) | **91.89** | **46.04** | **78.67** | **90.77** | **96.15** | **98.76** | **89.87** | 97.94 | **98.80** | **99.06** | **99.24** | **99.35** |
| Multi-Similarity + SEC (EMA, $\rho$=0.01, $\eta$=0.5/0.25) | 91.35 | 42.85 | 76.84 | 89.43 | 95.54 | 98.55 | 89.39 | **98.11** | 98.76 | **99.06** | 99.19 | 99.31 |

$^*$ We use "/" to separate the hyper-parameter settings for two datasets.

Table 6: Experimental results of face recognition. Face verification accuracy is reported on LFW, AgeDB30, and CFPFP while face identification accuracy is reported on MegaFace.

| Method | Face Verification | | | Size of MegaFace Distractors | | | |
|---|---|---|---|---|---|---|---|
| | LFW | AgeDB30 | CFPFP | $10^6$ | $10^5$ | $10^4$ | $10^3$ |
| Softmax | 98.97 | 91.30 | 93.39 | 80.43 | 87.11 | 92.83 | 96.12 |
| Sphereface [6] | 99.20 | 93.45 | 94.24 | 87.72 | 92.48 | 95.64 | 97.68 |
| Sphereface + $L^2$-reg ($\eta$=5e-3) | 99.28 | 93.42 | 94.30 | 88.38 | 92.86 | **95.93** | **97.87** |
| Sphereface + SEC ($\eta$=5.0) | 99.30 | 93.45 | 94.39 | 88.42 | 92.79 | 95.88 | 97.86 |
| Sphereface + SEC (EMA, $\rho$=0.4, $\eta$=1.0) | **99.33** | **94.02** | **94.93** | **88.74** | **92.91** | **95.93** | **97.87** |
| Cosface [7] | 99.37 | 93.82 | 94.46 | 90.71 | 94.30 | 96.57 | 98.09 |
| Cosface + $L^2$-reg ($\eta$=5e-3) | 99.12 | 94.32 | 94.64 | 91.03 | 94.46 | 96.85 | 98.24 |
| Cosface + SEC ($\eta$=5.0) | **99.42** | **94.37** | **94.93** | 91.13 | **94.63** | **96.92** | **98.37** |
| Cosface + SEC (EMA, $\rho$=0.4, $\eta$=1.0) | 99.18 | 94.17 | 94.79 | **91.31** | 94.61 | 96.85 | 98.33 |
| Arcface [8] | 99.22 | **94.18** | 94.69 | 90.31 | 94.07 | 96.67 | 98.20 |
| Arcface + $L^2$-reg ($\eta$=1e-3) | 99.32 | 93.93 | 94.77 | 90.68 | 94.34 | 96.83 | 98.32 |
| Arcface + SEC ($\eta$=5.0) | **99.35** | 93.82 | **94.91** | 90.91 | 94.56 | 96.95 | 98.37 |
| Arcface + SEC (EMA, $\rho$=0.4, $\eta$=1.0) | 99.27 | 93.90 | 94.74 | **91.02** | **94.74** | **97.02** | **98.46** |

Table 7: Experimental results of contrastive self-supervised learning with SimCLR [4]. Top 1/5 accuracy of linear evaluation is reported.

| Method | Training | CIFAR-10 | | CIFAR-100 | |
|---|---|---|---|---|---|
| | Epoch | Top 1 | Top 5 | Top 1 | Top 5 |
| NT-Xent [4] | 100 | 84.76 | 99.36 | 58.43 | 85.26 |
| NT-Xent + $L^2$-reg ($\eta$=5e-3/1e-3) | | 86.64 | 99.56 | 61.43 | 87.23 |
| NT-Xent + SEC ($\eta$=0.25/0.01) | | **86.87** | **99.64** | 61.66 | 87.33 |
| NT-Xent + SEC (EMA, $\rho$=0.2, $\eta$=0.5/0.05) | | 86.82 | 99.58 | **61.88** | **87.85** |
| NT-Xent [4] | 200 | 89.05 | 99.69 | 65.73 | 89.64 |
| NT-Xent + $L^2$-reg ($\eta$=5e-3/1e-3) | | 90.14 | 99.73 | 66.57 | 90.18 |
| NT-Xent + SEC ($\eta$=0.25/0.01) | | **90.35** | **99.77** | 66.25 | 90.12 |
| NT-Xent + SEC (EMA, $\rho$=0.2, $\eta$=1.0/0.05) | | 90.21 | 99.75 | **66.59** | **90.41** |

more comprehensive comparison between the original and this new version of SEC is shown below in the next subsection.

From Table 4, we also notice that different $\rho$ leads to slightly different improvements, where we suspect that $\rho$ would affect the magnitude of norms by $\mu_t$ and then further influence the performance. This observation implies that the magnitude of embedding norms would also influence the model optimization and the final result and thus $\rho$ may need further adjustments to achieve a higher improvement.

## D.2 Quantitative results on three tasks

Here we evaluate the new version of SEC (EMA) on three tasks as in the original paper. The implementation details are the same as in Appendix B, except that on face recognition, the value of $\eta$ at the $t$-th iteration is determined by

$$\eta_t = \min(\eta, \frac{500 * t}{\text{num. of total iterations}}).  \quad (11)$$

The results are shown in Table 5, 6, and 7, where we observe that SEC (EMA) further boosts the performance of the original SEC under several settings. In Table 5 of deep metric learning task,

where $\rho$ is not carefully tuned and simply set to 0.01, we observe that on CUB200-2011 dataset, SEC (EMA) improves the NMI, F1, and R@1 of multi-similarity loss with SEC by 1.37%, 2.6%, and 0.99%, respectively. On Cars196 dataset, SEC (EMA) also shows a remarkable improvements on NMI, F1, and R@1 of triplet loss with SEC by 2.08%, 3.36%, and 6.68%, respectively. In Table 6 of face recognition task, we observe that on MegaFace dataset with $10^6$ distractors, the rank-1 accuracies of sphereface, cosface, and arcface, with SEC, are enhanced by 0.32%, 0.18%, and 0.11%, respectively. In Table 7 of contrastive self-supervised learning task, on CIFAR-100 dataset, the top-1 linear evaluation accuracy of SimCLR with SEC is also improved by 0.22% and 0.34% when training for 100 and 200 epochs, with the help of SEC (EMA). These observations indicate the effectiveness of SEC (EMA) under some circumstances and it could be regarded as a more general version of SEC.