[Reviews · NeurIPS 2020]

Review 1

Summary and Contributions: The paper discusses deep metric learning methods that use L2 normalized embedding. They demonstrate the impact of the embedding norm by showing the effect on gradients with respect to cosine and d Euclidean distance losses. The authors claim that to further improve these methods, it is beneficial to regularize the embeddings to reside on the same sphere by adding a loss term to keep embedding's norm close to the average value across the batch. They continue to she that this modification improves performance of metric learning methods and reduces embedding norm variance.

Strengths: The work is written clearly, with good motivation and simple explanations. The main idea, of requiring the embedding to reside on same sphere, is intuitive yet novel (to best of my knowledge). Furthermore, the authors give empirical evidence that this may be an issue (showing the variance of norm in Figure 1) and later demonstrate how their method resolves it (Figure 4). The introduced method is clearly explained, with robustness measurements (with respect to hyperparameters) and shown to provide significant improvements in the metric learning task chosen. Overall, this work provides a novel method that can be easily incorporated to several tasks and domains with clear explanation and good evidence to support it.

Weaknesses: Some formulations may be a bit confusing, such as dealing with both cosine and distance losses of normalized embedding. As both are similar on normalized setting (up to an additive constant and scaling), I feel that using both of them in section 3 is unnecessary. I also think that authors should add applications and use cases beyond metric learning, as their method is applicable to other methods where representations are constrained to a sphere (see additional feedback part)

Correctness: I did not find any faults with claims or method. empirical methodology is on par with similar works.

Clarity: The paper is well written

Relation to Prior Work: The paper discusses relation to previous works in deep metric learning literature. It is missing some discussion concerning other L2norm-normalized methods such as contrastive learning for unsupervised/self-supervised learning.

Reproducibility: Yes

Additional Feedback: I think the authors can benefit from using their methods on other norm constrained methods. For example: "A Simple Framework for Contrastive Learning of Visual Representations" Chen et al. "Supervised Contrastive Learning" Khosla et al.


Review 2

Summary and Contributions: The paper introduces a new training method for deep metric learning. To measure the similarity between two feature embeddings, angular distance, which disentangles the norm and direction of embeddings, is widely adopted. However, most of the existing methods decouple the magnitude of each embedding by simply dividing it by its norm. The paper empirically and theoretically shows that the current normalization strategy can cause the instability of the update gradient in batch optimization and hinder the quick convergence of the embedding learning. The paper addresses this problem by adding a penalty term that directly regularizes the norm of embeddings to the original training objective. The paper claims that the proposed method can improve the baseline methods by a large margin on both deep metric learning and face recognition tasks.

Strengths: Soundness: The claim is well formulated mathematically, and the paper provides convincing empirical evidence (figure 4 and figure 5) to support the claim. Significance: One great advantage of the methods is that it is complementary to the other existing methods. Performance: The method consistently improves the performance on all the datasets by a significant margin.

Weaknesses: Soundness: The proposed method lacks algorithmic novelty. The idea of regularizing the \ell_2 norm of embedding vectors has already been proposed in [1] and [2]. It should be clearly stated how the proposed method differs from the previous regularization methods. Also, the paper states that the traditional angular loss functions result in unstable update gradient (line 137-147) but does not provide any empirical results on that. Performance: The paper does not provide a code for reproducibility. Also, hyper-parameters for training a network for deep metric learning task, such as learning rate and epoch, are not listed in the paper. Furthermore, it does not provide any references for selecting hyper-parameters. [1] Kihyuk Sohn, Improved Deep Metric Learning with Multi-class N-pair Loss Objective, NeurIPS 2016. [2] Hyun Oh Song, Stefanie Jegelka, Vivek Rathod, and Kevin Murphy, Deep Metric Learning via Facility Location, CVPR 2017.

Correctness: I checked the correctness of all the propositions.

Clarity: Yes. The paper is well written and easy to follow.

Relation to Prior Work: The paper does not mention some previous works on regularizing \ell_2 norm of embedding vectors, which are similar to the proposed method.

Reproducibility: No

Additional Feedback: Related works: Please write a brief description of each prior work, instead of just listing the names (line 65-67). Hyper-parameters: Are hyper-parameters borrowed from the original papers or tuned by grid search? Please mention how to choose hyper-parameters and network structures clearly. Performance: For an ablation study, please conduct an experiment with \mu=0 instead of average embedding norm and show how the proposed method differs from the previous regularization methods. Also, the phrase 'normalized n-pair loss' is somewhat ambiguous. Does it stand for 'Tuplet Margin Loss' in [3]? If not, you should include [3] as a baseline. [3] Baosheng Yu and Dacheng Tao, Deep Metric Learning with Tuplet Margin Loss, ICCV 2019 ------- Thank you for doing new experiments and making changes to take into account the feedback. However, I disagree with your claim that the original N-pair loss uses inner product without l2-normalization (see Section 3.2.2 in the original paper). So, I leave my rating unchanged.


Review 3

Summary and Contributions: In this paper, the authors have proposed a deep metric learning method with spherical embedding. Existing angular distance based methods ignore the norms of the learned features, which may lead to unstable gradient in batch optimization. This paper analyzes the effect of the embedding norms and also proposes a spherical embedding constraint (SEC) to minimize it. Experimental results show the effectiveness of the proposed method.

Strengths: 1) The problem of the embedding norms of angular based methods is interesting, which is ignored by most existing works. 2) The paper is easy to read. 3) Extensive experiments on various applications.

Weaknesses: 1) I think the main drawback of this paper is that, the design of the SEC loss is somehow too straightforward and trivial. It simply minimizes the distance between the norm of each feature and the average norm, which acts as an additional term of the existing losses. A more elegant design should be expected for the NeurIPS level. 2) The reviewer is not clear whether \mu is the average norm *of the batch*, or *of all the features*? It seems the latter, but in this case how to update \mu during training? Whenever the parameters of the metric are updated, all the features are changed, and \mu should be re-calculated. This detail is very important for the paper. 3) An important baseline that should be compared with is that, after the last layer we use an additional layer to normalize the output feature (e.g., to make the norms as 1 for all the features). Then, we use the normalized features as the output instead of the original ones, where the losses operate on the normalized features. It's required to show the advantages of the proposed SEC over this intuitive design. 4) In the experiments, we observe that some methods have relatively large improvement after using SEC, while some do not. The reviewer wants to know if the larger improvements in accuracy are coming from the larger reduction of the variance of feature norms.

Correctness: Yes

Clarity: Yes

Relation to Prior Work: Yes

Reproducibility: No

Additional Feedback: The rebuttal partly addresses my concerns. However, the reviewer still considers that the technical contributions are not significant enough, and also the experiments need improvement. Thus the reviewer keeps the original rating of the paper.


Review 4

Summary and Contributions: This paper proposes a regularizer to encourage the underlying embeddings (before normalization) to have similar norms. Based on the training dynamics of SGD, this results in more balance updates on the angles between the embeddings.

Strengths: It is shown that the proposed method can be used to improve many existing methods such as those in [4, 10, 17, 11].

Weaknesses: The results shown in the analysis/theory part of this work are known. Proposition 1 was shown in Wang et al Deep Metric Learning with Angular Loss, Section 3, figure 3. Proportion 2 was shown in Zhang et al Heated-Up Softmax Embedding, Section 3.3 The motivation of this work is based on vanilla SGD. It is unclear whether the problem exists with other widely used optimizers. In particular, the problems stated may not be true for Adam (the one used in the experiments). In Adam, the gradient is weighted by an exponential moving average of each parameter, and it may make the magnitude of the gradient similar regardless of the underlying norm. Some more analysis is needed especially because the experiments are done with Adam.

Correctness: The claims are correct but known.

Clarity: Yes.

Relation to Prior Work: I am wondering about the interplay of the proposed method and batch norm/ weight norm. These method can make the embedding norms close too. Is Figure 1 generated with batch norm?

Reproducibility: Yes

Additional Feedback: [I have read the authors' feedback and respectfully disagree with the authors that the results are novel in comparison with the two previous works. I did not see a reply to the SGD vs. other optimizer concern I raised.]

[Author Response · NeurIPS 2020]

We thank the four reviewers for their constructive comments. The following are our responses to reviewers' comments. (We use T, SHT, NNP, and MS to denote triplet, semihard triplet, normalized n-pair, and multi-similarity, respectively.)

**To Reviewer #2 Q1**: Some formulations are confusing. **R1**: Thanks. We will rewrite the formulations in the revision.

**Q2**: More applications beyond metric learning? More discussions? **R2**: We add experiments of SimCLR on CIFAR10 in Table 1. The classifier is trained with lr in {2, 5, 10} and bs=256 for 50 epochs. We will also discuss more related works accordingly, e.g., contrastive learning in unsupervised/self-supervised learning as suggested by the reviewer.

**To Reviewer #3 Q1**: Difference with previous regularization methods? One more ablation study? **R1**: (1) Motivation: The original N-pair loss uses inner product without $l_2$-normalization as the similarity measure, which aims to optimize only the direction and remove the influence of norms. However, we consider losses with $l_2$-normalization to alleviate the unbalanced direction update caused by large norm variance. (2) Formulation: The $l_2$-regularizer in N-pair loss constrains the norms to be small, while SEC reduces the norm variance and is more effective as shown in Table 2. Besides, for the method Clustering, it only uses the common $l_2$-normalization.

**Q2**: Empirical results of unstable update gradient? **R2**: Since it's difficult to directly show the gradient, in Figure 1 we provide the unstable change of samples' norms, which are the denominator factors of gradient magnitudes of corresponding embeddings, to reflect this instability.

**Q3**: Hyper-parameters and how to determine them? code? **R3**: The training settings are in Table 3. The hyper-parameters in losses follow [1] (Section 5) for T, the original paper for SHT, [2] for NNP (we test s=25 and 64), and original authors' GitHub for MS. Other settings such as network structure are following the paper of MS. We will release the code once this paper is accepted.

**Q4**: A brief description of each prior work. **R4**: We will rewrite more detailly in the revision.

**Q5**: Normalized n-pair loss? **R5**: Thanks. It actually stands for Equation 3 in our paper. We add experiments of tuplet margin loss (TML) (w.o. and with SEC) in Table 4. We use hyper-parameters ($\beta = 0.1, \lambda = 0.5, \epsilon = 0.01$) in the original paper, bs=128 (4 instances/class), and Adam.

**Q6**: More broader impact discussions. **R6**: We will add more discussions in the revision.

**To Reviewer #4 Q1**: The straightforward and trivial design of SEC. **R1**: Thanks. Though the formulation is straightforward, the underlying goal of SEC is not trivial, aiming to adjust the gradient contributions from different embeddings. In particular, we introduce a novel perspective of the impact of large norm variance for angular loss optimization, which offers an important guidance for the related algorithm design both theoretically and empirically. Further, compared to another $l_2$-regularizer, the experiments show that SEC is a better choice (please see Reviewer #3' R1 for details) and is useful for many different kinds of angular losses.

**Q2**: The calculation of average norm. **R2**: Thanks. In practice, we only calculate the average norm in a mini-batch. From Figure 2, we observe that the averaged norm is smoothly changing and finally stable.

**Q3**: Compare with an intuitive baseline? **R3**: The baseline losses in Table 3 and 4 in our paper have already operated on the $l_2$-normalized features and SEC is designed to reduce the norm variance of embeddings when using $l_2$-normalization.

**Q4**: Larger improvements come from larger variance reduction? **R4**: The variance reduction on Cars training set: 5.77→0.02 for T, 2.82→0.03 for SHT, 1.76→0.13 for NNP, and 1.68→0.004 for MS, thus this conclusion makes sense.

**Q5**: More broader impact discussions. **R5**: We will add more discussions in the revision.

**To Reviewer #5 Q1**: Part of the analysis/theory are known. **R1**: Thank you for the comment, however, for Proposition 1, we believe that the Section 3 and Figure 3 in [3] haven't show that $\frac{\partial L}{\partial f}$ is vertical to $f$. For Proposition 2, we agree that the Section 3.3 in [4] also mentions that the magnitude of the gradient is inversely proportional to the embedding norm (we will add it to related works), however, we take a further step by explaining how this gradient influences the direction update and how to solve the problem, which are ignored by [4].

**Q2**: More analysis about the optimizer? **R2**: Thanks. Adam (and other optimizers with adaptive lr) will adjust the lr for each model parameter according to its historical gradient magnitude, resulting in current gradient magnitude changed. We think it helps balance the update of each parameter in some extent. However, for each individual parameter, Adam would not further analyze its gradient compositions from different embeddings and separately adjust these components considering the influence of different embedding norms. Therefore, we suspect that Adam would alleviate this problem to some extent, but the unbalanced direction update among embeddings caused by large norm variance still exists.

**Q3**: Interplay of SEC and batch norm/weight norm? **R3**: Figure 1 is generated without BN on top of the final embedding and we add two contrast experiments: (1) adding BN on top of the final embedding before $l_2$-normalization (2) employing weight normalization for the final fc layer. We use SHT on Cars and the results are shown in Table 5. We observe that BN/WN may not help reduce the norm variance and the added BN does harm to SEC.

**Reference**: [1] Song et al. Deep Metric Learning via Lifted Structured Feature Embedding. CVPR 2016. [2] Yu et al. Deep Metric Learning with Tuplet Margin Loss. ICCV 2019. [3] Wang et al. Deep Metric Learning with Angular Loss. ICCV 2017. [4] Zhang et al. Heated-Up Softmax Embedding.

Table 1: SimCLR with SEC (ResNet50 encoder, linear head, NT-Xent, dim=128, bs=256, temp=0.5, SGD. Best accuracy of the classifier is reported).

| Method | Epoch | LR | Top 1 |
|---|---|---|---|
| SimCLR (w.o. SEC) | 150 | cosine decay (0.1-0.5 grid search) | 86.63 |
| SimCLR (with 0.1*SEC) | 150 | cosine decay (0.1-0.5 grid search) | **87.00** |

Table 2: Comparisons of $l_2$-norm regularizers (Cars, SHT).

| η | μ | NMI | F1 | R@1 | Mean | Var. |
|---|---|---|---|---|---|---|
| 0 | - | 67.64 | 38.31 | 80.17 | 7.63 | 2.82 |
| 0.5 | avg | 72.67 | 44.67 | 85.19 | 3.20 | 0.03 |
| 0.1 | 0 | 64.57 | 34.72 | 75.78 | 0.45 | 0.01 |
| 0.01 | 0 | 68.05 | 38.49 | 80.61 | 1.41 | 0.07 |
| 0.005 | 0 | 69.24 | 40.24 | 82.60 | 1.98 | 0.14 |
| 0.001 | 0 | 69.13 | 40.62 | 81.54 | 4.36 | 0.79 |

Figure 1: The changing curves of several samples' norms.

Table 3: Hyper-parameters.

| Dataset | Iters | Loss | [lr for backbone]/[lr for head]/[lr decay @iter] |
|---|---|---|---|
| CUB | 8k | T, SHT | 5e-6/2.5e-6/0.1@5k |
| | | NNP | 1e-5/5e-6/0.1@5k |
| | | MS | 5e-5/2.5e-5/0.1@5k |
| Cars | 8k | T, SHT, NNP | 1e-5/1e-5/0.5@4k, 6k |
| | | MS | 4e-5/4e-5/0.1@2k |
| SOP, In-shop | 12k | T, SHT, NNP, MS | 5e-4/1e-4/0.1@6k |

Table 4: TML with SEC.

| Dataset | LR | Method | NMI | F1 | R@1 |
|---|---|---|---|---|---|
| CUB | 4e-5/2e-5/ 0.1@5k | TML | 68.96 | 39.25 | 63.64 |
| | | +SEC | **71.00** | **42.28** | **64.74** |
| Cars | 6e-5/6e-5/ 0.5@4k, 6k | TML | 69.78 | 41.12 | 82.87 |
| | | +SEC | **72.77** | **43.03** | **84.20** |
| SOP | 5e-4/1e-4/ 0.1@6k | TML | **90.50** | **38.45** | 73.66 |
| | | +SEC | 90.45 | 37.92 | **74.34** |
| | | | R@1 | R@2 | R@3 |
| In-shop | 1e-3/2e-4/ 0.1@6k | TML | 84.50 | 96.95 | 98.09 |
| | | +SEC | **84.74** | **97.38** | **98.27** |

Figure 2: The average norm in a mini-batch at each iteration.

Table 5: SEC with BN/WN.

| Method | NMI | F1 | R@1 | Mean | Var. |
|---|---|---|---|---|---|
| SEC | **72.67** | **44.67** | **85.19** | 3.20 | 0.03 |
| BN | 67.09 | 37.31 | 80.20 | 7.77 | 3.00 |
| BN+SEC | 47.71 | 15.71 | 62.46 | 7.01 | 0.03 |
| WN | 66.98 | 37.47 | 79.42 | 9.31 | 4.95 |
| WN+SEC | 71.81 | 43.10 | 85.01 | 3.18 | 0.03 |

[Meta-Review · NeurIPS 2020]

This paper points out a widespread problem with angular losses, and proposes a simple, elegant scheme to address the problem (regularizing each embedding to lie on a shell), getting moderate but consistent improvements across a range of problem settings and datasets. As pointed out by Reviewer 5, the majority of the theoretical results were already known in Section 3.3 of "Heated-Up Softmax Embedding" (2018, unpublished, https://arxiv.org/abs/1809.04157 ). That paper, however, did not really propose a solution to the problem, merely noted its existence. Reviewer 5 also complains that the interaction with the Adam optimizer is under-explored in this work. Reviewer 3 points out that previous work, e.g. "Improved Deep Metric Learning with Multi-class N-pair Loss Objective," also regularized the L2 norm of embedding vectors (towards 0; see their Section 3.2.2). Table 2 in your rebuttal, however, shows clearly that this yields substantially worse performance than your proposal of regularizing towards the average, on one particular problem. Reviewer 4's primary remaining complaint after the rebuttal is that the proposed SEC scheme is too simple. In this case, where there are dozens of papers that might have benefited from this scheme but (so far as any of us know) have not invented it, I don't view this as a disadvantage, but rather a benefit, since it will be trivially easy for future work using angular loss functions to employ SEC. Thus, I am left to agree with Reviewer 2, who views this work as "a novel method that can be easily incorporated to several tasks and domains with clear explanation and good evidence to support it," and therefore recommend acceptance. Some changes are necessary in the camera-ready version: - First, you need to cite "Heated-Up Softmax Embedding" in your theory section (and introduction) as having made most of the same observations in the past. The novel contribution of this paper to the literature, then, is primarily the SEC method and the demonstration that it works reliably in several settings. - The various changes to the discussion, clarifications, additional experiments, etc. mentioned in the rebuttal will all improve the paper. In particular, some more discussion – and, ideally, experimentation – of the situation with the common Adam optimizer would be helpful. (Incidentally, I also found the terminology "vertical to" quite unusual; the standard term would be "perpendicular" or "orthogonal.") - In addition, I would strongly recommend adding at least a subset of the mu=0 methods considered in the rebuttal's Table 2 to all of your experiments in the paper, to more convincingly demonstrate that regularizing towards the mean norm is much superior to regularizing towards zero in a variety of settings. - Finally, one of the main application areas for SEC is in facial recognition, a (rightfully) extremely controversial area – see, e.g., https://dl.acm.org/doi/10.1145/3313129 or https://en.wikipedia.org/wiki/Facial_recognition_system#Controversies . This is not by any means grounds to reject your paper. But, seeing a paper significantly about facial recognition claim that discussions of broader impacts on society and ethical considerations are "not applicable for our work" reduces public trust that machine learning researchers are responsible members of society, who, say, can be trusted to operate without onerous regulation. You must update the Broader Impacts section to note that, although facial recognition is quite controversial as a technology, there is no reason to expect that SEC's mild improvement to facial recognition performance should make any substantial difference to its societal application, nor is it expected to exacerbate its e.g. racial unbalances. (In fact, it would be interesting to study, though perhaps beyond the scope of this paper, whether the unregularized embedding norms end up correlated to various demographic attributes, and hence SEC might even improve fairness in these systems.)